# TiTok: Transfer Token-level Knowledge via Contrastive Excess to Transplant LoRA

**Chanjoo Jung[1], Jaehyung Kim[1]**
[1]Yonsei University
{chanjoo0427,jaehyungk}@yonsei.ac.kr

## ABSTRACT

Large Language Models (LLMs) are widely applied in real world scenarios, yet fine-tuning them comes with significant computational and storage costs. Parameter-Efficient Fine-Tuning (PEFT) methods such as LoRA mitigate these costs; however, the adapted parameters are dependent on the base model and cannot be transferred across different backbones. One way to address this issue is through knowledge distillation, but its effectiveness inherently depends on training data. Recent work such as TransLoRA avoids this by generating synthetic data; nevertheless, this adds complexity since it requires training an additional discriminator model. In this paper, we propose **TiTok**, a new framework that enables effective LoRA **T**ransplanta**t**ion through **Tok**en-level knowledge transfer. Specifically, TiTok captures task-relevant information through a token-wise contrastive excess between a source model with and without LoRA. This excess highlights informative tokens and enables selective filtering of synthetic data, all without additional models or overhead. Through experiments on three benchmarks across multiple transfer settings, we demonstrate that TiTok is consistently effective, achieving average performance gains of + 4–10% compared to baselines overall.[1]

## 1 INTRODUCTION

Large Language Models (LLMs) (Brown et al., 2020; Vaswani et al., 2023) have made significant progress in many real-world applications, including chatbots (OpenAI et al., 2024), search engines (Xiong et al., 2024), and coding assistants (Rozière et al., 2024). While fine-tuning LLMs has been demonstrated to be a promising way to improve performance on downstream tasks, it incurs substantial computational and storage costs. To alleviate this, Parameter-Efficient Fine-Tuning (PEFT) (Houlsby et al., 2019) methods such as LoRA (Hu et al., 2021) update only a small subset of parameters while keeping the base model frozen. However, PEFT's adapted parameters are dependent to the base model and thus cannot be transferred across different models. This limitation is increasingly critical given the rapid release of new LLMs and the growing diversity of available models.

One potential approach to mitigate this limitation is Knowledge distillation (KD) (Hinton et al., 2015; Azimi et al., 2024), which transfers the knowledge embedded in a source model's PEFT adapters to a target model with new PEFT adapters by aligning the target's output distributions with those of the source. However, KD is inherently data-dependent, typically requiring access to training data from target downstream tasks (Nayak et al., 2019; Liu et al., 2024), which is often unavailable or costly to obtain. To address this limitation, TransLoRA (Wang et al., 2024) has recently proposed to use synthetic data, leveraging the data synthesis capabilities of recent LLMs (Wang et al., 2023; Kim et al., 2025). This approach enables the target model to acquire domain knowledge without direct access to the original dataset. Nevertheless, TransLoRA requires training an additional discriminator model to filter low-quality synthetic data, which inevitably introduces extra complexity and computational overhead. Furthermore, it primarily emphasizes the role of synthetic data, paying less attention to *how the knowledge transfer process itself should be designed*.

**Contribution.** In this paper, we propose a new framework that enables effective LoRA **T**ransplanta**t**ion through **Tok**en-level knowledge transfer (**TiTok**). Our high-level idea is to se-

---

[1]Code available at https://github.com/NaughtyMaltiz16/TiTok

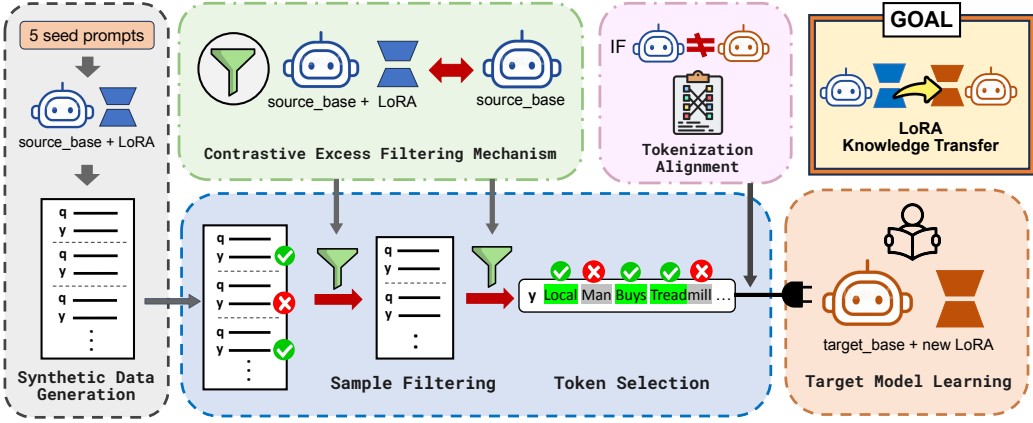

Figure 1: **Overview of TɪTᴏᴋ**: **T**ransplantation through **Tok**en-level knowledge transfer. Starting from a small set of seed prompts, the source expert model (source base model + LoRA) generates synthetic data. A token-wise contrastive excess filtering mechanism then compares the expert against its base backbone to compute token-level excess scores. Using these scores, TɪTᴏᴋ first performs sample filtering and subsequently token selection, retaining only the most informative samples and tokens. When tokenizers differ, masks are aligned prior to training. The resulting filtered data is finally used to train a new LoRA on the target backbone, enabling efficient knowledge transfer.

lectively convey task-relevant information from the source model's LoRA by using token-level signals to guide the transfer process, rather than relying on the entire token sequence. We specifically capture this information through a concept we introduce as *token-wise contrastive excess*, which is obtained by comparing predictions from a source model with LoRA and the same model without LoRA. Intuitively, this token-wise contrastive excess highlights tokens that contain important task knowledge. This excess signal is further utilized to filter the generated synthetic data for training, thereby enabling selective learning on samples that contain richer information. Unlike TransLoRA, which requires training an additional discriminator model for filtering, TɪTᴏᴋ requires neither extra models nor additional training overhead. Moreover, we design an effective mechanism to resolve tokenizer mismatches between source and target models, enhancing robustness and applicability.

We demonstrate the effectiveness of TɪTᴏᴋ by conducting extensive experiments on three widely used benchmarks, which cover both reasoning (Big Bench Hard (Suzgun et al., 2022) and MMLU (Hendrycks et al., 2021)) and personalization tasks (LaMP (Salemi et al., 2024)). In particular, TɪTᴏᴋ improves the performance by +9.94% over the vanilla target model, +8.5% over KD, and +4.4% over TransLoRA, when averaged across all tasks and transfer settings. We also explored a variety of transfer settings, including transfers within the same model, across different model families and sizes, and even across different model versions. In every case, our approach achieves consistent improvements. Interestingly, TɪTᴏᴋ remains effective even when applied to external data originating from tasks different from the target task, highlighting both robustness and general applicability. Overall, these empirical results highlight TɪTᴏᴋ as a methodologically simple yet powerful paradigm for efficiently transferring LoRA knowledge across models in diverse scenarios.

## 2 RELATED WORKS

**Transferring PEFT adapters.** Parameter-Efficient Fine-Tuning (PEFT) (Houlsby et al., 2019; Li & Liang, 2021) has emerged as both a practical and popular alternative to full model fine-tuning. By requiring updates to only a small portion of parameters, it enables efficient adaptation, with LoRA (Low-Rank Adaptation) (Hu et al., 2021; Dettmers et al., 2023) standing out as one of the most widely adopted methods. However, a fundamental limitation is that LoRA adapters are tied to the frozen backbone they were trained on, making them difficult to transfer to other base models. To address this issue, recent studies such as TransLoRA (Wang et al., 2024) have attempted to

transplant the knowledge of LoRA adapters across models by generating synthetic data (Wang et al., 2023). While effective to some extent, this approach requires an additional discriminator model to filter high-quality synthetic data, resulting in a relatively heavy pipeline. In parallel, Knowledge Distillation (KD) (Hinton et al., 2015; Azimi et al., 2024) has been widely explored as another means of knowledge transfer; however, traditional KD typically operates within a teacher–student framework at the logit or sequence level and requires access to training data close to the original, in order to use the teacher's distribution for supervising the student. In contrast, our approach focuses on **token-level selective transfer**, offering a more fine-grained yet lightweight alternative that enables efficient transplantation of LoRA adapters across diverse models for deployment.

**Data synthesis with LLMs.** Synthetic data generation with LLMs has attracted increasing attention as a means to reduce reliance on costly or inaccessible datasets (Wang et al., 2023; Kim et al., 2025). Prior lines of research have leveraged synthetic data for purposes such as privacy preservation (Bu et al., 2025), data augmentation (Kumar et al., 2021), and domain adaptation (Li et al., 2023). In our framework, synthetic data serves as a core component, enabling the transfer of LoRA adapters without access to the original training corpus, while simultaneously mitigating privacy concerns and reducing the degree of dependency on external datasets. Moreover, since both queries and labels are generated directly by the source expert model itself, synthetic data ultimately provides a self-sufficient mechanism that fits to our objective of lightweight and effective knowledge transfer.

**Selective token training.** Recent studies (Lin et al., 2025; Gu et al., 2020) demonstrate that not all tokens contribute equally to model training, motivating research on selective training strategies. While such approaches have primarily been applied to accelerate optimization or reduce redundancy (Yeongbin et al., 2024; Bal et al., 2025), our work innovatively extends this concept to the setting of knowledge transfer. Specifically, our concept of excess scores (Eq. 2) is derived from this idea, where the contrast between the source backbone and its LoRA adapter yields token-level judgments. This enables TITOK to transplant LoRA knowledge in a more focused and fine-grained manner, highlighting the broader applicability of selective training beyond its original scope.

## 3 TITOK: TRANSPLANTING LORA THROUGH TOKEN-LEVEL KNOWLEDGE

In this section, we introduce TITOK, a framework for LoRA **T**ransplanta**ti**on through **Tok**en-level knowledge transfer (Fig. 1). The core idea of TITOK is to transfer the knowledge from a source model's LoRA adapter into a target model's LoRA adapter by training selectively on the informative tokens within synthetic data. Specifically, the framework consists of three components: (1) *Synthetic data generation* (Sec. 3.1), where the source expert model produces query–label pairs for target task; (2) *Excess score computation* (Sec. 3.2), which calculates token-level importance using the source model; and (3) *Target model training with filtering* (Sec. 3.3), which trains the target model with newly initialized LoRA adapter on top-ranked samples and tokens. In addition, we propose *Excess score alignment* (Sec. 3.4), an algorithm designed to apply TITOK even when tokenizers of the source and target models differ. The overall algorithm of TITOK is presented in Alg. 1.

### 3.1 SYNTHETIC DATA GENERATION VIA LLM PROMPTING WITH FEW-SHOT DATA

Let $\mathcal{M}_s$ denote the source backbone LLM and $\mathcal{A}_s$ its LoRA adapter on target downstream task, which forms the source expert model $\mathcal{M}_s + \mathcal{A}_s$. The target model, whose LoRA adapter $\mathcal{A}_t$ will be trained, is denoted by $\mathcal{M}_t$. Then, TITOK first constructs a synthetic dataset $\mathcal{D}_s$, similar to the idea of TransLoRA (Wang et al., 2024). This usage of synthetic data allows us to avoid keeping the whole original dataset for the downstream task, and simultaneously let $\mathcal{A}_t$ learn the knowledge encoded in the source adapter $\mathcal{A}_s$. Unlike TransLoRA's approach of using the untuned target model $\mathcal{M}_t$ to generate synthetic data, we use the source expert model $\mathcal{M}_s + \mathcal{A}_s$ to synthesize data (see empirical comparison in Fig. 3). Concretely, $\mathcal{M}_s + \mathcal{A}_s$ synthesizes both the *query* and the *label* within a prompting-based data synthesis framework (Wang et al., 2023) (see details in Appendix P); given a few-shot data of downstream task, it first generates a query $\mathbf{q}$, and then produces the corresponding label $\mathbf{y}$ conditioned on $\mathbf{q}$. To encourage diversity, we apply ROUGE-L filtering together with deduplication to all tasks, except for exceptional cases where such filtering is infeasible (more details are in Appendix I). Consequently, the resulting synthetic dataset consists of query–label pairs:

$$\mathcal{D}_s = \{(\mathbf{q}_j, \mathbf{y}_j)\}_{j=1}^N. \tag{1}$$

In this way, $\mathcal{M}_t$ would be trained on the synthetic data $\mathcal{D}_s$ generated by the source expert model $\mathcal{M}_s + \mathcal{A}_s$, thereby enabling knowledge transfer without relying on the entire original dataset.

## 3.2 TOKEN-WISE CONTRASTIVE EXCESS SCORE FROM SOURCE MODEL WITH LORA

As described in Sec. 3.1, TITOK relies on synthetic data, but synthetic data is often prone to imperfections (Chen et al., 2024b); thus, sophisticated filtering is essential to retain high-quality informative samples. TransLoRA (Wang et al., 2024) tackles this challenge with a separate discriminator to filter useful queries, but this introduces the extra burden of training and maintaining an extra model. In contrast, we propose a lightweight alternative that only utilizes the already trained source model. Specifically, we use source model and its LoRA adapter to perform two complementary roles: (1) the *amateur* role ($\mathcal{M}_s$) and (2) the *expert* role ($\mathcal{M}_s + \mathcal{A}_s$). Then, the difference between the two roles provides an *implicit supervision signal* where the task information is encoded.

Formally, let $\mathbf{y} = [y_1, ..., y_L]$ denote the synthesized response, where $y_i$ is a token of $\mathbf{y}$ corresponding to the synthesized query $\mathbf{q}$. Then, we define the *excess score* as:

$$S(y_i) = L_e(y_i) - L_a(y_i), \tag{2}$$

where the amateur and expert losses on token $y_i$ are defined as:

$$L_a(y_i) = \log P_{\mathcal{M}s}(y_i \mid \mathbf{q}, \mathbf{y}{<}i), \quad L_e(y_i) = \log P_{\mathcal{M}_s+\mathcal{A}s}(y_i \mid \mathbf{q}, \mathbf{y}{<}i). \tag{3}$$

The excess score $S(y_i)$ quantifies the knowledge discrepancy introduced by the LoRA adapter, thereby identifying the tokens where the adapter provides a decisive contribution. Intuitively, if the backbone model is uncertain about predicting a token but the LoRA-enhanced model assigns it with high confidence, that token will thus obtain a large excess score. This implies that tokens with higher $S(y_i)$ correspond to positions where the LoRA adapter injects task-specific knowledge that the backbone could not capture on its own. In this way, the excess score functions as a *fine-grained attribution signal*, derived entirely from the internal behavior of the model itself, and guides training toward the specific regions of data that are most enriched with the adapter's knowledge.

While the excess score is motivated from an intuitive contrast between the backbone and its LoRA-enhanced counterpart, it is also grounded in principled statistical theory. Theoretically, the proposed token-wise contrastive excess score is connected to a token-level log-likelihood ratio (LLR) between the source expert (backbone + LoRA) and the backbone. LLR is a robust metric widely established in statistical testing (Li & Babu, 2019). For example, by the Neyman–Pearson lemma (Neyman & Pearson, 1933), the likelihood ratio is known to be the optimal statistic for identifying differences between two models. Therefore, high-LLR tokens are exactly the regions where the adapter changes the predictive distribution, *i.e.*, where task knowledge is injected. In addition, from the perspective of standard knowledge distillation, tokens with nearly zero LLR provide no additional teacher knowledge and offer no benefit for transfer. In contrast, high-LLR tokens correspond to maximal teacher–student divergence, larger gradients, and higher information contribution to the student. Therefore, emphasizing tokens with high LLR in particular naturally focuses training on the precise positions where transferable knowledge is concentrated.

## 3.3 TARGET MODEL TRAINING WITH SAMPLE FILTERING AND TOKEN SELECTION

After the computation of excess scores $S(y_i)$, the newly initialized LoRA adapter $\mathcal{A}_t$ for target model $\mathcal{M}_t$ is trained using the synthetic samples $(\mathbf{q}_j, \mathbf{y}_j) \in \mathcal{D}_f$ with two-level filtering schemes.

**First stage: Sample filtering.** We begin by filtering the synthetic dataset $\mathcal{D}_s$ (Eq. 1) at the sample level to remove less informative examples. For each synthetic sample, we compute the mean of the excess scores $S(y_i)$ across the tokens in $\mathbf{y}$ and retain only $M$ samples with the highest values.

$$\bar{S}_j = \frac{1}{|\mathbf{y}_j|} \sum_{y_i \in \mathbf{y}_j} S(y_i). \tag{4}$$

Let $\mathcal{D}_f$ be the set of the $M$ samples in $\mathcal{D}_s$ with the largest $\bar{S}_j$:

$$\mathcal{D}_f = \text{Top}M\big\{ (\mathbf{q}_j, \mathbf{y}_j) \in \mathcal{D}_s : \bar{S}_j \big\}. \tag{5}$$

Through this step, the synthetic data undergoes a filtering process, ensuring that subsequent training is concentrated specifically on the remaining examples in $\mathcal{D}_f$ with richer knowledge signals.

---

**Algorithm 1:** TITOK: Transplanting LoRA through Token-Level Knowledge

---

**Input:** source expert $\mathcal{M}_s + \mathcal{A}_s$, target $\mathcal{M}_t$, parameters $N, M, k\%$
**Output:** trained target LoRA $\mathcal{A}_t$

1. Construct synthetic dataset $\mathcal{D}_s = \{(\mathbf{q}_j, \mathbf{y}_j)\}_{j=1}^N$ with $\mathcal{M}_s + \mathcal{A}_s$
2. **for** $(\mathbf{q}_j, \mathbf{y}_j) \in \mathcal{D}_s$ **do**
$\quad$ Compute token excess scores $S(y_i) = L_e(y_i) - L_a(y_i)$
$\quad$ Calculate mean score $\bar{S}_j = \frac{1}{|\mathbf{y}_j|} \sum_{y_i \in \mathbf{y}_j} S(y_i)$
3. Select top-$M$ samples by $\bar{S}_j$ to form $\mathcal{D}_f$
4. **for** $(\mathbf{q}_j, \mathbf{y}_j) \in \mathcal{D}_f$ **do**
$\quad$ Rank tokens by $S(y_i)$ and keep top-$k\%$, represented by mask $I_{k\%}(y_i)$
5. **if** *tokenizer*$(\mathcal{M}_s) \neq$ *tokenizer*$(\mathcal{M}_t)$ **then**
$\quad$ Align masks $I_{k\%}^{(s)}(y_i) \rightarrow I_{k\%}^{(t)}(y_i)$
6. Train $\mathcal{A}_t$ on $\mathcal{M}_t$ with masked loss $\mathcal{L}_{\texttt{TiTok}} = \sum_{(\mathbf{q}_j, \mathbf{y}_j) \in \mathcal{D}_f} \sum_{y_i \in \mathbf{y}_j} I_{k\%}(y_i) \cdot L_t(y_i)$

**return** $\mathcal{A}_t$

---

**Second stage: Token selection.** Next, we consider token selection; that is, $\mathcal{A}_t$ does not learn from all tokens within the retained samples. Instead, it focuses only on those prioritized by the excess scores $S(y_i)$, which are identified as most important for knowledge transfer. To achieve this, we select the top $k\%$ of tokens ranked by their excess scores using the indicator $I_{k\%}(y_i)$:

$$I_{k\%}(y_i) = \begin{cases} 1, & \text{if } \text{rank}_{\mathbf{y}_j}(S(y_i)) \leq \lfloor k\% \cdot |\mathbf{y}_j| \rfloor, \\ 0, & \text{otherwise,} \end{cases} \tag{6}$$

where $|\mathbf{y}_j|$ denotes the number of tokens in $\mathbf{y}_j$, and $\text{rank}_{\mathbf{y}_j}(S(y_i))$ indicates the rank of $S(y_i)$ among the tokens of that response. Based on this selection, the training objective for $\mathcal{A}_t$ is defined as

$$\mathcal{L}_{\texttt{TiTok}} = \sum_{(\mathbf{q}_j, \mathbf{y}_j) \in \mathcal{D}_f} \sum_{y_i \in \mathbf{y}_j} I_{k\%}(y_i) \cdot L_t(y_i), \tag{7}$$

where $L_t(y_i)$ is the negative log-likelihood loss assigned by $\mathcal{M}_t + \mathcal{A}_t$ on token $y_i$ (only $\mathcal{A}_t$ is learnable). By training only on these filtered tokens (Eq. 7), TITOK enables $\mathcal{A}_t$ to efficiently acquire the source LoRA's knowledge without access to the original training data or any external models.

### 3.4 EXCESS SCORE ALIGNMENT ACROSS DIFFERENT TOKENIZERS

In cases of transfer between models with different tokenizers, a direct mapping of token-level signals is not possible, given that the source and target models may segment texts differently. To address this, we introduce a simple yet robust tokenizer alignment algorithm that propagates token masks (Eq. 6) from the source token sequence $\mathbf{y}^{(s)}$ to the target token sequence $\mathbf{y}^{(t)}$. The algorithm first aligns token sequences using dual pointers that incrementally decode and match text spans. Masks are then propagated using the following four rules: (1) direct copy for one-to-one mappings, (2) replication for one-to-many, (3) averaging for many-to-one, and (4) averaging with replication for many-to-many. Finally, a top-$k\%$ selection step retains the most confident target tokens. This process ensures consistent supervision across tokenizers, enabling reliable transfer even when models tokenize text differently. The conceptual illustration of this procedure is presented in Fig.4.

## 4 EXPERIMENT

In this section, we present our experimental results to answer the following research questions:

- **RQ1:** Can TITOK efficiently transfer knowledge of LoRA in various scenarios? (Table 1)
- **RQ2:** What is the contribution of each component in TITOK? (Table 2)
- **RQ3:** How sensitive is token-level selective transfer to the selection ratio? (Figure 2)
- **RQ4:** How does the choice of model to synthesize query affect the performance? (Figure 3)
- **RQ5:** Can TITOK transfer knowledge using data from a different or unrelated domain? (Table 4)

## 4.1 EXPERIMENTAL SETUPS

**Models.** We mainly present our knowledge transfer experiments using models from the Mistral and Llama families. Specifically, we designed the following LoRA transfer (*source → target*) setups. (1) Mistral-7B-Inst-v0.3[2] → Mistral-7B-Inst-v0.3: the basic transfer setup, (2) Mistral-7B-Inst-v0.3 → Llama-3.1-8B-Inst[3]: the different-family model transfer setup, (3) Llama-3.2-3B-Inst[4] → Llama-3.1-8B-Inst: the different-size model transfer setup, and (4) Llama-2-7b-chat-hf[5] → Llama-3.1-8B-Inst: the different-version model transfer setup. These setups are intended to test whether a smaller model can effectively transfer knowledge to a larger one, and to explore whether a relatively weaker model can still influence a newer, stronger model. These various setups realistically reflect how LLMs develop today, where various models are released and newer, improved versions keep emerging. Model names are abbreviated for clarity and are used consistently in all tables and figures: 1) "Mistral 7B" = Mistral-7B-Instruct-v0.3, 2) "Llama3 8B" = Llama-3.1-8B-Instruct, 3) "Llama3 3B" = Llama-3.2-3B-Instruct, 4) "Llama2 7B" = Llama-2-7b-chat-hf.

**Baselines.** To demonstrate the effectiveness of TITOK, we compare it against three baselines: (i) *Vanilla*, the target base model without any fine-tuning; (ii) *KD(+MinED)*, knowledge distillation (KD) from source expert model (Hinton et al., 2015). When the source and target models use different tokenizers, the original KD is not applicable. In this case, we use Minimum-Edit-Distance (MinED) tokenizer alignment (Wan et al., 2024), which aligns both token sequences and vocabulary distributions via dynamic-programming sequence alignment for near matches (e.g., "gets" ↔ "get") and by mapping probability mass to nearest edit-distance neighbors (e.g., "immediately" ↔ "immediate"). We use the synthesized data by TransLoRA as the training data for KD (+MinED) ; and (iii) *TransLoRA* (Wang et al., 2024), a prior method in which the vanilla target model synthesizes the queries, while the source model with its LoRA adapter generates the corresponding synthetic labels. A discriminator is then employed to filter the synthetic data for training the target LoRA adapter.

**Datasets.** Following prior work in TransLoRA, we first conduct experiments on two representative benchmarks: (1) *Big-Bench Hard (BBH)* (Suzgun et al., 2022) and (2) *Massive Multitask Language Understanding (MMLU)* (Hendrycks et al., 2021). BBH consists of 27 challenging reasoning tasks structured as multiple choice or short answer questions, designed to test compositional generalization and advanced problem solving abilities of a Language Model. Meanwhile, MMLU covers 57 tasks across diverse academic subjects, presented in multiple choice format to evaluate broad knowledge and reasoning skills of a model. As both benchmarks provide only test sets, we construct 90%/10% train-evaluation splits. For BBH, we use the full dataset, whereas for MMLU, we randomly sample 100 instances of the dataset for each task before applying the 90%/10% split.

To extend our approach to personalization and text generation, we additionally conduct experiments on LaMP benchmark (Salemi et al., 2024), focusing exclusively on its generation tasks. In particular, we experiment with *(3) News Headline Generation (LaMP 4)* and *(4) Scholarly Title Generation (LaMP 5)*, as they are the only text generation tasks that are both accessible and reliably evaluable. The remaining LaMP tasks are excluded, given that LaMP 1,2,3 are discriminative, and LaMP 6,7 lack gold labels. For LaMP tasks, the source expert model $\mathcal{M}_s + \mathcal{A}_s$ is trained on data from the 30 users with the longest activity histories. From each user, we use 200 data points for training and 50 data points for validation, a design choice intended to conduct a more rigorous and robust evaluation. In total, each LaMP task contains 6,000 training examples and 1,500 evaluation examples.

To assess performance on the BBH and MMLU, we measure the average accuracy using the LM-Eval Harness (Gao et al., 2024). Following the setup used in TransLoRA (Wang et al., 2024), all tasks are conducted in a zero-shot setting. Meanwhile, for the LaMP tasks, we adopt ROUGE-1 and ROUGE-L scores as evaluation metrics, following the benchmark's primary evaluation metrics.

**Implementation details.** When training both the source and target models, we use a learning rate of $5 \times 10^{-5}$, train for 2 epochs with a batch size of 4, and apply LoRA with rank $r = 8$, scaling factor $\alpha = 8$, and dropout 0.05. Optimization is performed using AdamW with weight decay

---

[2] https://huggingface.co/mistralai/Mistral-7B-Instruct-v0.3
[3] https://huggingface.co/meta-llama/Llama-3.1-8B-Instruct
[4] https://huggingface.co/meta-llama/Llama-3.2-3B-Instruct
[5] https://huggingface.co/meta-llama/Llama-2-7b-chat-hf

Table 1: **Main results.** Experiments on BBH, MMLU, News Headline and Scholarly Title Generation tasks under four transfer settings. Reasoning tasks (BBH, MMLU) are evaluated via LM-Eval Harness, and personalization tasks (News Headline, Scholarly Title Generation) via ROUGE-1/L. Zero-shot evaluations are reported as the mean across 3 random seeds. Best scores are in **bold**.

| Transfer | Method | BBH | MMLU | News Headline | | Scholarly Title | |
|---|---|---|---|---|---|---|---|
| | | Acc. | Acc. | ROUGE-1 | ROUGE-L | ROUGE-1 | ROUGE-L |
| Mistral 7B → Mistral 7B | Vanilla | 0.397 | 0.557 | 0.117 | 0.101 | 0.381 | 0.311 |
| | KD (+MinED) | 0.417 | 0.560 | 0.117 | 0.104 | 0.385 | 0.310 |
| | TransLoRA | 0.416 | 0.534 | 0.156 | 0.137 | 0.447 | 0.382 |
| | **TɪTᴏᴋ (ours)** | **0.424** | **0.561** | **0.161** | **0.143** | **0.473** | **0.413** |
| Mistral 7B → Llama3 8B | Vanilla | 0.469 | 0.469 | 0.125 | 0.110 | 0.444 | 0.378 |
| | KD (+MinED) | 0.475 | 0.482 | 0.127 | 0.112 | 0.454 | 0.387 |
| | TransLoRA | 0.473 | 0.473 | 0.126 | 0.110 | 0.461 | 0.397 |
| | **TɪTᴏᴋ (ours)** | **0.484** | **0.485** | **0.139** | **0.123** | **0.464** | **0.403** |
| Llama3 3B → Llama3 8B | Vanilla | 0.469 | 0.469 | 0.125 | 0.110 | 0.444 | 0.378 |
| | KD (+MinED) | 0.474 | 0.477 | 0.125 | 0.110 | 0.449 | 0.383 |
| | TransLoRA | 0.471 | 0.467 | 0.122 | 0.108 | 0.454 | 0.387 |
| | **TɪTᴏᴋ (ours)** | **0.496** | **0.478** | **0.127** | **0.113** | **0.456** | **0.392** |
| Llama2 7B → Llama3 8B | Vanilla | 0.469 | 0.469 | 0.125 | 0.110 | 0.444 | 0.378 |
| | KD (+MinED) | 0.473 | 0.476 | 0.125 | 0.110 | 0.449 | 0.382 |
| | TransLoRA | 0.472 | 0.468 | 0.123 | 0.109 | 0.457 | 0.394 |
| | **TɪTᴏᴋ (ours)** | **0.488** | **0.477** | **0.138** | **0.120** | **0.461** | **0.403** |

$1 \times 10^{-2}$, together with a linear learning rate schedule and a warmup ratio of $0.1$. For synthetic data generation, we provide five samples from the original training data as few-shot exemplars, and apply top-p sampling (Holtzman et al., 2020) to generate both queries and labels, with sampling hyperparameters tuned individually for each task. To further encourage diversity, we apply ROUGE-L filtering with a threshold of $0.7$ and deduplication to remove redundant queries, following Wang et al. (2023). For tasks where ROUGE-based filtering is infeasible, we apply only deduplication (see Appendix I). For the initial synthetic pool, we generate $2M$ synthetic samples and, after filtering with token-wise contrastive excess scores, retain the top $M$, where $M$ equals the source training set size (see Appendix H). For token selection, the selection ratio $k\%$ is fixed at $70\%$ across all tasks and transfer settings, except for the Llama3 3B → Llama3 8B transfer setting, where we find that $k\% = 30\%$ consistently yields the best performance. During inference in the evaluation stage, we adopt greedy decoding to ensure fully deterministic and reproducible inference results.

## 4.2 MAIN RESULTS

Table 1 summarizes the experimental results on BBH, MMLU, and LaMP (News Headline and Scholarly Title generation) tasks across four transfer settings. First, we observe that transfer within the same model family is highly effective; when transplanting the LoRA adapter trained on the Mistral 7B into a fresh instance of the same model, TɪTᴏᴋ consistently surpasses all baselines. In particular, TɪTᴏᴋ improves the vanilla model by 23.94% on average across all tasks, and further outperforms the KD and TransLoRA baselines by 21.95% and 4.75%, respectively. Taken together, the findings show that, as a first step, transfer within the same family is reliably successful.

Beyond same family transfer, we also find that TɪTᴏᴋ is highly effective in cross-model transfer settings. For example, when transferring from Mistral 7B to Llama 8B, TɪTᴏᴋ delivers an average improvement across all tasks of 6.79% over the vanilla model, while outperforming KD by 4.69% and TransLoRA by 4.86%. This highlights that TɪTᴏᴋ is not confined to intra-family knowledge transfer, but can successfully bridge across architectures. In the case of Llama3 3B to Llama3 8B, TɪTᴏᴋ yields average gains of 3.07% against vanilla, 2.18% against KD, and 3.02% against TransLoRA. These results suggest that TɪTᴏᴋ scales effectively with model size, making it useful when moving from lightweight to larger models without losing efficiency. Finally, when transferring from Llama2 7B to Llama3 8B, TɪTᴏᴋ performs average advantages of 5.95% over vanilla, 5.17% over KD, and 5.13% over TransLoRA. This indicates that TɪTᴏᴋ adapts robustly even across different model versions, implying practical relevance when models are upgraded in real-world pipelines.

Table 2: **Ablation study.** "Sample filtering" uses the mean token-wise contrastive excess score to remove uninformative examples, while "Token selection" further refines the retained data by selectively keeping only the most informative tokens within each sample. Without sample filtering, data are randomly sampled. Results are reported as accuracy (Acc.) for BBH and MMLU, averaged across tasks. Meanwhile, we report ROUGE-1 (R-1) and ROUGE-L (R-L) for LaMP tasks.

| Settings | | BBH | MMLU | News Headline | | Scholarly Title | |
|---|---|---|---|---|---|---|---|
| Sample filtering | Token selection | Acc. | Acc. | R-1 | R-L | R-1 | R-L |
| ✗ | ✗ | 0.458 | 0.485 | 0.133 | 0.117 | 0.456 | 0.393 |
| ✗ | ✓ | 0.463 | 0.496 | 0.137 | 0.121 | 0.460 | 0.397 |
| ✓ | ✗ | 0.470 | 0.500 | 0.139 | 0.122 | 0.460 | 0.397 |
| ✓ | ✓ | **0.483** | **0.501** | **0.142** | **0.125** | **0.464** | **0.403** |

Table 3: **Excess-score token subgroup analysis.** Effect of token subgroup selection in Mistral 7B → Llama3 8B transfer. The top 20% excess-score tokens yield the best performance on BBH.

| Transfer | Method | Subgroup | Acc. |
|---|---|---|---|
| | Vanilla | – | 0.469 |
| Mistral 7B → Llama3 8B | | **top 20%** | **0.482** |
| | TITOK | bottom 20% | 0.468 |
| | | random 20% | 0.476 |

Collectively, these results provide strong evidence that TITOK not only excels in transfers within the same model family, but also demonstrates broad effectiveness in cross-model transfers, thereby highlighting its robustness across a wide spectrum of model scales, versions, and families.

## 4.3 ADDITIONAL ANALYSES

In the following sections, we present additional analyses to showcase the effectiveness of TITOK. Unlike the main table which reports the average of three random seeds to ensure statistical stability, the empirical results discussed in the subsequent sections are reported using a single, fixed seed.

**Ablation study.** To validate the contribution of each component in our framework, we conduct additional experiments by selectively excluding the sample filtering and token selection mechanisms in Sec. 3.3. We report the average performance scores across all four transfer settings, and the results are presented in Table 2. When sample filtering is not applied (1st and 2nd rows), the training data for the target model is randomly sampled from the full synthetic dataset. Under this setting, applying only token selection (2nd row) performs noticeably better than the purely random baseline. This demonstrates the effectiveness of our token selection approach and empirically confirms that token-wise contrastive excess reliably identifies and selects the most informative tokens. Similarly, when only sample filtering is applied (1st and 3rd rows), the results improve over pure random sampling, indicating that selecting high-quality data is essential. This finding further implies that incorporating an effective filtering mechanism for synthetic data is essential. In our framework, token-wise contrastive excess serves this role by reliably selecting samples with richer and more informative signals, as demonstrated clearly by the empirical results. Finally, the results show that combining both stages (4th row) achieves the best performance, confirming their complementary roles in filtering high-quality examples and selecting informative tokens for effective knowledge transfer.

Next, we also evaluate whether high excess-score tokens indeed carry concentrated transferable knowledge. We verify this empirically by contrasting token subgroups of the top 20%, bottom 20%, and random 20% on BBH in the Mistral 7B → Llama3 8B transfer setting. To prevent unintentional overlap across groups and to extract the most informative region, we employ a comparatively low 20% threshold for this analysis. The results are presented in Table 3. Analytically, the empirical results show that the top 20% produces the best transfer performance, validating tat tokens with high token-wise contrastive excess scores indeed contain concentrated task knowledge.

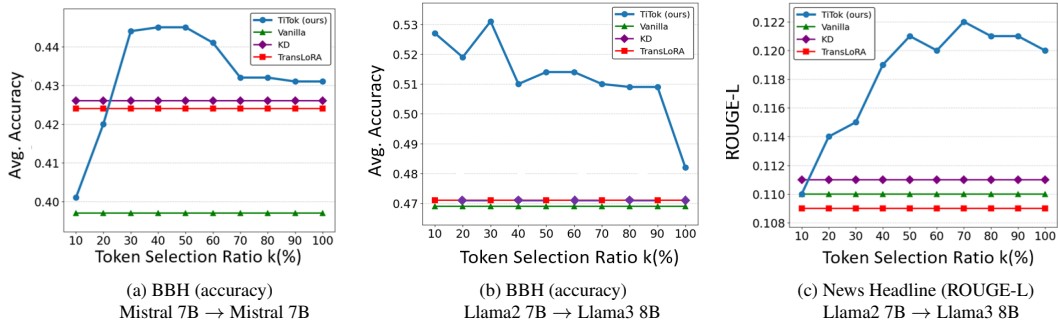

Figure 2: **Representative performance trends across $k\%$.** Among the three graphs, two come from the same task (BBH), while two share the same transfer setting (Llama2 7B → Llama3 8B).

Interestingly, we observe that truly important tokens are consistently selected regardless of the expert's model. For example, for BBH datasets, Mistral 7B and Llama2 7B experts agree on 59.76% of the chosen tokens even when we limit the selection to a small top 20% subset to prevent trivial overlap. This implies that a token will continue to be recognized across models if it is truly significant for the task, and it can be effectively identified by our token-wise contrastive excess score.

**Effect of token selection ratio.** We now proceed to explore the impact of token selection ratios ($k\%$) in our framework. Fig. 2 presents three representative curves: one task (BBH) under two transfer settings, and another task (News Headline Generation) under the same transfer setting for comparison. For BBH with the same backbone on source and target (Mistral 7B→Mistral 7B), a moderate token selection ratio of 30-70% yields the best results. Very low ratios (10-20%) lead to underfitting, while a 100% ratio is ineffective as it applies no filtering at all. Interestingly, in a weak-to-strong transfer (Llama2 7B → Llama3 8B) for the same BBH task, performance improves as $k\%$ decreases. This suggests that selecting only the tokens with the highest excess loss effectively filters out the majority of noisy regions where the weaker model is uncertain. By discarding the majority of tokens where the weaker model lacks confidence, TITOK can significantly reduce negative transfer. This trend, however, reverses for the News Headline Generation task under the same weak-to-strong transfer setting. For this task, a larger $k\%$ is generally better, potentially because even a weaker model can provide valuable lexical and stylistic cues that are useful for personalization. Consequently, unlike reasoning-focused tasks such as BBH, which are sensitive to noisy supervision, personalization tasks like News Headline Generation benefit more broadly from the source model's outputs, even when the source model is relatively weaker than the target model.

**Impact of query generation model.** We now examine how the choice of query generation model influences performance. Practically, synthetic queries can be generated either by the source model or by the target model, with the latter being the approach originally adopted in TransLoRA's pipeline (Wang et al., 2024). Figure 3 compares these two options across different transfer settings, with the reported scores averaged over all BBH tasks. Notably, we observe that using the source expert model (source backbone + LoRA) for query generation generally yields stronger performance than using the target model. One possible explanation is that when synthetic data is generated by the same model, they remain closer to its training distribution. This alignment between queries and labels likely provides more coherent supervision, thereby facilitat-

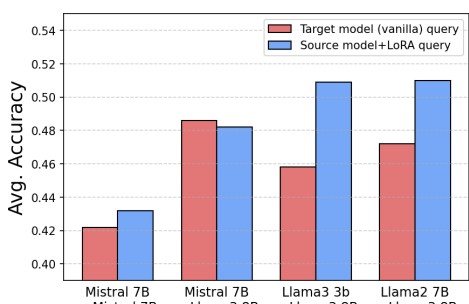

Figure 3: **Impact of query source.** Across transfer settings, using the source expert model to synthesize query generally yields better performance. The accuracy averaged over all BBH tasks are reported.

ing more effective transfer. Ultimately, these findings empirically justify our design choice to use the source model as both the query and label generator, instead of using the target model.

**Effectiveness of transfer through external data source.** We further examine whether TITOK can transfer knowledge effectively in cases where synthetic data is not preferred, and thus external data is used as an alternative. To this end, we evaluate three alternative settings on LaMP tasks for Mistral 7B→Mistral 7B transfer setup: (1) using data from a randomly chosen different user, (2) mixing data from multiple users, and (3) transferring across tasks, where data of Scholarly Title Generation is used to train on News Headline Generation and vice versa (*i.e.*, out-of-distribution scenario). The results are presented in Table 4. Remarkably, TITOK consistently outperforms all baselines across these

Table 4: **Transfer using external data.** ROUGE-1 (R-1) and ROUGE-L (R-L) on News Headline and Scholarly Title Generation under three data settings for Mistral 7B→Mistral 7B: (1) other-user, (2) mixed-user, (3) cross-task. Best scores are highlighted in **bold**.

| Data Setting | Method | News Headline | | Scholarly Title | |
|---|---|---|---|---|---|
| | | R-1 | R-L | R-1 | R-L |
| Other-user | Vanilla | 0.117 | 0.101 | 0.381 | 0.311 |
| | KD | 0.127 | 0.113 | 0.405 | 0.333 |
| | **TITOK (ours)** | **0.151** | **0.136** | **0.481** | **0.425** |
| Mixed-user | Vanilla | 0.117 | 0.101 | 0.381 | 0.311 |
| | MinED | 0.124 | 0.110 | 0.409 | 0.337 |
| | **TITOK (ours)** | **0.151** | **0.137** | **0.480** | **0.422** |
| Cross-task | Vanilla | 0.117 | 0.101 | 0.381 | 0.311 |
| | MinED | 0.118 | 0.106 | 0.403 | 0.331 |
| | **TITOK (ours)** | **0.133** | **0.120** | **0.450** | **0.383** |

heterogeneous external settings. These findings demonstrate that TITOK is not restricted to synthetic data scenarios, but can also adapt effectively under external or user-provided data conditions. This highlights the flexibility of TITOK for practical deployment in diverse real-world application scenarios and further underscores its adaptability across different data conditions, as it is effective even when external data is used as an alternative to synthetic data.

**Transfer under cross-architecture and large-scale.** We additionally conduct experiments on larger-scale models and architectures to reinforce TITOK's effectiveness. Due to the large model size, all experiments in this section are evaluated on BBH using 4-bit quantization. As shown in Table 5, TITOK successfully transfers token-level knowledge in both settings. Specifically, cross-architecture transfer from a dense model (Mistral 7B) to a mixture-of-experts (MoE) model yields a 1.98% improvement over the target baseline,

Table 5: **Larger scale transfer settings.** BBH evaluation results for cross-architecture and large-scale model transfers.

| Transfer Setting | Method | Acc. |
|---|---|---|
| Mistral 7B → Mixtral-8×7B[6] | Vanilla | 0.454 |
| | **TITOK (k=70%)** | **0.463** |
| Llama-3.3-70B → Llama-3.3-70B[7] | Vanilla | 0.593 |
| | **TITOK (k=70%)** | **0.621** |

while scaling up to the 70B transfer setting, achieves a 4.72% gain. These results confirm that TITOK generalizes robustly across both complex MoE architectures and massive model scales.

# 5 CONCLUSION

In this paper, we propose TITOK, a novel and efficient framework that transfers LoRA knowledge from a source model to a target model by training only on a selectively chosen set of highly informative tokens. At its core, TITOK strategically leverages token-level signals to distill task-relevant information from the source adapter, rather than indiscriminately relying on the entire token sequence. With this simple yet effective design, TITOK consistently surpasses baselines in various transfer settings, providing a practical solution for efficient knowledge transfer.

**Limitations and Future Work** While TITOK has shown clear advantages, opportunities remain to refine the framework. First, generating synthetic data still requires a few seed examples. Nevertheless, this reliance is modest, as we avoid using full datasets. In addition, our experiments show external data serves as an effective alternative, reinforcing the flexibility of TITOK. Regarding token selection, while TITOK currently relies on a simple yet highly effective fixed threshold for token selection, future work could explore adaptive thresholding strategies to further enhance efficiency.

---

[6]https://huggingface.co/mistralai/Mixtral-8x7B-Instruct-v0.1
[7]https://huggingface.co/meta-llama/Llama-3.3-70B-Instruct

ETHICS STATEMENT

We conduct this research in full accordance with established ethical standards. For our experiments we rely exclusively on publicly available datasets such as LaMP, MMLU, and BBH and use them strictly in accordance with their intended purpose for academic research. For the LaMP tasks, which involve user data, our TᴇTᴏᴋ framework aligns with ethical considerations by minimizing data dependence. It does not store or expose raw user data and only updates a small set of task and user-specific parameters. This helps safeguard privacy while enabling efficient knowledge transfer.

REPRODUCIBILITY STATEMENT

We provide a comprehensive description of our implementation in Section 4, including pipeline configurations, hyperparameters, models, datasets, and evaluation metrics. The source code of TᴇTᴏᴋ is publicly available at https://github.com/NaughtyMaltiz16/TiTok.

ACKNOWLEDGMENTS

This research was supported in part by Institute for Information & communications Technology Planning & Evaluation (IITP) grant funded by the Korea government (MSIT) (No. RS-2020-II201361, Artificial Intelligence Graduate School Program (Yonsei University); No. RS-2025-25442405, Development of a Self-Learning World Model-Based AGI System for Hyperspectral Imaging).

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

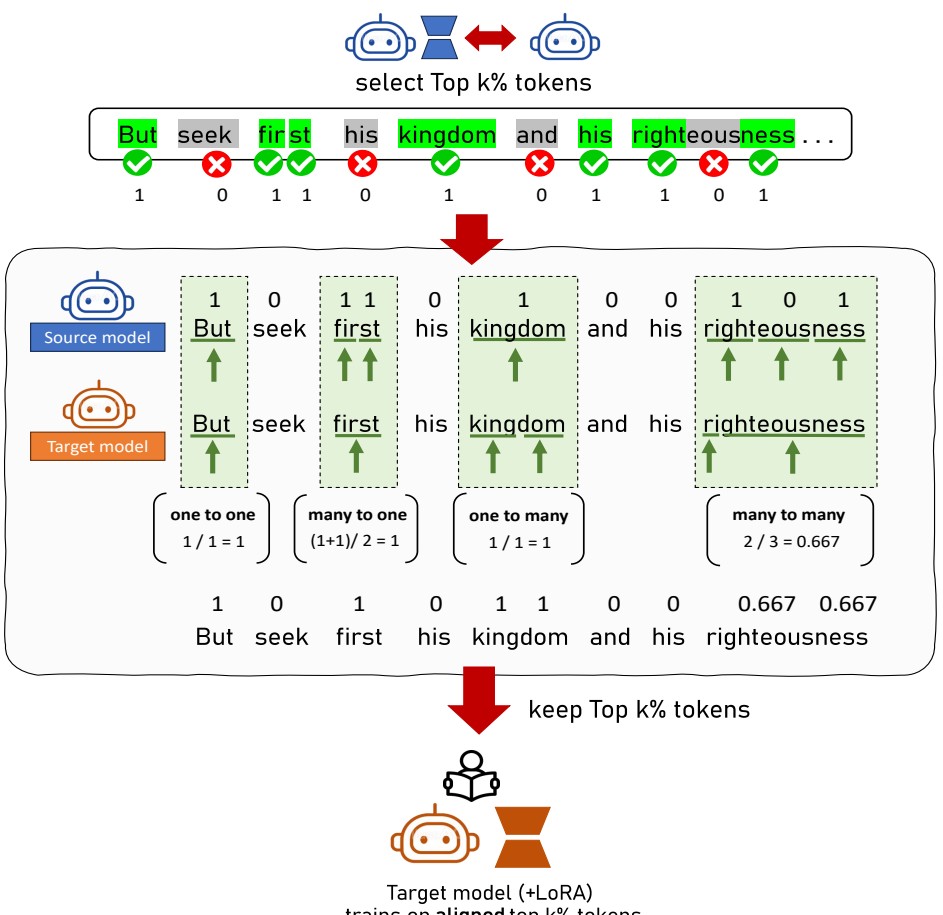

Figure 4: **Overview of TITOK's tokenizer alignment algorithm.** The algorithm handles cases where the source and target models use different tokenizers. The binary mask scores assigned by the source model are averaged within aligned spans and propagated to target tokens, producing fractional scores that guide top-$k\%$ token selection for training the target model's LoRA adapter.

## A  TOKENIZER ALIGNMENT ALGORITHM

When the source and target models use different tokenizers, direct token-level transfer is not possible. To address this, we implement a dual-pointer alignment procedure. The algorithm maintains two pointers, one for the source tokens and one for the target tokens. At each step, the source pointer advances by one token, accumulating a decoded segment, while the target pointer incrementally extends its own segment until the normalized texts match. Once a match is found, the corresponding spans are recorded as an alignment, and the target pointer jumps forward to that position. After alignment, we apply masking rules to propagate the source binary mask scores (values that indicate whether a token should be kept or discarded) to the target tokens. Specifically:

1) **One-to-One**: binary mask score is directly copied.

2) **One-to-Many**: score is replicated across all aligned target tokens.

3) **Many-to-One**: averaged scores of multiple source tokens are assigned to the target token.

4) **Many-to-Many**: the averaged score of the source tokens is assigned to the corresponding aligned target tokens.

Table 6: Average alignment percentage breakdown across BBH and News Headline tasks in the Mistral 7B → Llama3 8B transfer setting.

|  | BBH | News Headline |
|---|---|---|
| exceptions | 0% | 0% |
| many to many | 46.18% | 5.00% |
| many to one | 3.47% | 17.45% |
| one to many | 0.01% | 0.01% |
| one to one | 50.33% | 77.52% |

Table 7: Effect of token alignment cases in the Mistral-7B → Llama-3 8B transfer experiments.

|  |  | BBH | News Headline | |
|---|---|---|---|---|
|  |  | Acc. | R-1 | R-L |
| | Vanilla | 0.469 | 0.125 | 0.110 |
| Mistral-7B → Llama-3 8B | One to one only | 0.472 | 0.138 | 0.120 |
| | TITOK (k=70%) | 0.482 | 0.160 | 0.142 |

This process yields fractional mask scores that capture the relative importance of each target token. Finally, we keep only the top-$k\%$ of tokens according to these scores, producing a final binary selection mask over the target tokens that preserves the most informative regions while discarding less relevant ones. An overview of this tokenizer alignment algorithm is presented in Figure 4.

**Robustness of the algorithm**. The algorithm is conceptually error-free since it simply performs a deterministic mapping between two tokenizers on the exact same text sequence. Since both tokenizations correspond to the identical underlying character string, the mapping cannot introduce semantic errors; every token in both tokenizers is defined over non-overlapping spans of the same text, and these spans align uniquely. For instance, in the Mistral 7B → Llama3 8B transfer setting on BBH and News Headline benchmarks, 100% of tokens are aligned, and there are no exception cases as shown in Table 6, confirming that alignment errors do not occur in practice.

Furthermore, we additionally construct a degraded setting in which we keep only the one-to-one aligned token pairs and discard all other alignment cases. We perform the experiment in the Mistral 7B → Llama3 8B transfer setting on BBH and News Headline tasks, and the full results are presented in Table 7. From these empirical results, we can observe that using all the full alignment cases outperforms using one-to-one token pairs only. This implies that the many-to-one, one-to-many, and many-to-many alignments are also correctly aligned, as including them yields the best results. Together, these results confirm that our alignment method is both accurate and robust.

## B USAGE OF AI ASSISTANTS

AI assistants are minimally used in this work, restricted solely to language refinement, such as grammar correction, punctuation, and sentence structure. All research ideas, methodologies, and analyses are the original contributions of the authors. Thus, the use of AI is confined to editorial support and did not influence the originality or intellectual contributions of the work.

## C DETAILS OF DATASETS

### C.1 BIG-BENCH HARD (BBH)

Big-Bench Hard (BBH) (Suzgun et al., 2022) is designed as a rigorous benchmark for evaluating model performance on challenging reasoning problems, including multi-step logical reasoning, symbolic manipulation, and commonsense inference. The tasks are formatted as multiple choice questions. Since BBH is originally a test-only benchmark, we split 90% of the data for training the source expert model and reserved 10% for evaluation. The 27 BBH tasks are categorized in Table 8.

## C.2 Massive Multitask Language Understanding (MMLU)

Massive Multitask Language Understanding (MMLU) (Hendrycks et al., 2021) is a comprehensive benchmark for evaluating model performance across a broad range of knowledge intensive tasks. The benchmark consists of multiple choice questions and, similarly to BBH, we also split the 100 randomly sampled test-only instances into 90%/10%. All the 57 subtasks are categorized in Table 9.

Table 8: Categorization of the 27 Big-Bench Hard (BBH) tasks.

| Category | Tasks |
|---|---|
| Logical Reasoning | boolean_expressions, causal_judgement, date_understanding, disambiguation_qa, dyck_languages, formal_fallacies, logical_deduction_three_objects, logical_deduction_five_objects, logical_deduction_seven_objects, temporal_sequences, tracking_shuffled_objects_three_objects, tracking_shuffled_objects_five_objects, tracking_shuffled_objects_seven_objects, web_of_lies |
| Linguistic | hyperbaton, ruin_names, salient_translation_error_detection, snarks, word_sorting |
| Mathematical / Symbolic | geometric_shapes, multistep_arithmetic_two, object_counting, reasoning_about_colored_objects |
| Applied / Knowledge | movie_recommendation, navigate, penguins_in_a_table, sports_understanding |

Table 9: Categorization of the 57 MMLU tasks.

| Category | Tasks |
|---|---|
| STEM | abstract_algebra, anatomy, astronomy, college_biology, college_chemistry, college_computer_science, college_mathematics, college_medicine, college_physics, computer_security, conceptual_physics, electrical_engineering, elementary_mathematics, formal_logic, high_school_biology, high_school_chemistry, high_school_computer_science, high_school_mathematics, high_school_physics, high_school_statistics, machine_learning, medical_genetics, nutrition, professional_medicine, professional_psychology, virology, clinical_knowledge, human_aging, human_sexuality |
| Humanities | business_ethics, formal_fallacies, jurisprudence, logical_fallacies, philosophy, prehistory, world_religions, moral_disputes, moral_scenarios, professional_law, professional_accounting, high_school_world_history, high_school_us_history, high_school_european_history, global_facts, security_studies |
| Social Sciences | econometrics, high_school_macroeconomics, high_school_microeconomics, management, marketing, public_relations, sociology, us_foreign_policy, high_school_geography, high_school_government_and_politics |
| Other | miscellaneous, global_facts |

## C.3 LaMP tasks

We utilize the LaMP benchmark to evaluate whether TiTok is also effective in the personalization setting. Among the LaMP tasks, we focus on the two text generation tasks that are suitable, accessible, and evaluable in our setting:

- **News Headline Generation (LaMP 4).** Given an author profile consisting of previously written headlines, the model is asked to generate a headline for a news article. The task evaluates if the model can adapt its output to reflect the author's characteristic style in journalistic writing.

- **Scholarly Title Generation (LaMP 5).** Using an author profile built from prior titles of academic publications, the model generates a title for a new given abstract. The task assesses the model's ability to capture and reproduce distinctive conventions of academic writing.

# D    BASELINE DETAILS

To evaluate the effectiveness of TITOK, we compare against several baselines as follows:

## D.1    VANILLA

The vanilla baseline corresponds to the standard target base model without any additional training or knowledge transfer from the source model. This setup reflects the performance of the target model in its raw initialized state and serves as a lower bound for evaluating transfer methods. Intuitively, achieving performance that surpasses this baseline provides clear evidence that the target model has successfully acquired and internalized knowledge transferred from the source model.

## D.2    KNOWLEDGE DISTILLATION (KD) (+MINED)

The knowledge distillation (KD) (Hinton et al., 2015; Azimi et al., 2024) baseline trains the student model to mimic the teacher model's output distribution. Specifically, the loss is a weighted sum of the cross entropy objective and the KL divergence between the teacher and student distributions. In our setup, the KD experiments are conducted using the TransLoRA filtered synthetic datasets.

For cases where the source and target models use mismatched tokenizers when performing KD, we apply the Minimal Edit Distance (MinED) alignment method (Wan et al., 2024). In particular, MinED matches tokens across different vocabularies by minimizing the number of character level edits. For example, it can align "get" with "gets," "color" with "colour," or "analysis" with "analyses." This tokenizer alignment approach avoids degeneracy issues in token alignment when conducting KD, though it is different from our dual-pointer based alignment algorithm.

## D.3    TRANSLORA

The TransLoRA baseline (Wang et al., 2024) transfers LoRA knowledge by generating synthetic data. In this method, the vanilla target model synthesizes queries, and the source model with its LoRA adapter provides the corresponding labels. Subsequently, a discriminator, separately trained with the source model as its base, is applied to filter the synthetic data used for fine-tuning the target model's LoRA adapter. For consistency and fair comparison, we use the same hyperparameter settings for synthetic data generation in our experiments .

**Comparison with TransLoRA**. TransLoRA's core assumption is to transfer entire synthetic sequences generated by the teacher. Therefore, its framework is tightly coupled to synthetic data and is unable to function in the absence of teacher-generated labels. In that sense, this also implies that TransLoRA cannot be applied to external datasets. Thus, it mainly highlights the role of synthetic data, giving less thought to the architecture of the knowledge transfer procedure itself.

In contrast, TITOK directly rethinks the mechanism of knowledge transfer itself. TITOK concentrates on determining which tokens actually contain expert-specific knowledge (i.e., the tokens where the expert significantly deviates from the basic model) rather than copying entire synthetic sequences. Our novel token-level contrastive signal fundamentally changes how knowledge is extracted and aligned. Because this process does not rely on teacher-generated labels, TITOK naturally extends to external, non-synthetic datasets (See Table 4) and is therefore much more generalizable.

Therefore, despite the fact that the ultimate objective of both approaches is to solve the same high-level LoRA transfer problem, which inevitably leads to certain overlapping components, the underlying philosophies are essentially different. TransLoRA focuses on transferring full outputs and relies entirely on synthetic data, whereas TITOK focuses on transferring the expert's internal knowledge signals regardless of the data source, thus being more generalizable and effective.

Table 10: **Comparison with a few-shot KD baseline using five seed prompts.** BBH and MMLU are reported as the average accuracy across tasks, while News Headline and Scholarly Title Generation tasks are evaluated using ROUGE-1 (R-1) and ROUGE-L (R-L). *KD (5-shot)* denotes knowledge distillation performed with only five few-shot examples. Best scores are in **bold**.

| Transfer | Method | BBH Acc. | MMLU Acc. | News Headline R-1 | News Headline R-L | Scholarly Title R-1 | Scholarly Title R-L |
|---|---|---|---|---|---|---|---|
| Mistral 7B → Mistral 7B | Vanilla | 0.397 | 0.557 | 0.117 | 0.101 | 0.381 | 0.311 |
| | KD (5-shot) | 0.402 | 0.558 | 0.118 | 0.104 | 0.383 | 0.312 |
| | TITOK (ours) | **0.432** | **0.563** | **0.160** | **0.142** | **0.473** | **0.414** |
| Mistral 7B → Llama 8B | Vanilla | 0.469 | 0.469 | 0.125 | 0.110 | 0.444 | 0.378 |
| | KD (5-shot) | 0.470 | 0.477 | 0.126 | 0.111 | 0.446 | 0.379 |
| | TITOK (ours) | **0.482** | **0.488** | **0.140** | **0.124** | **0.464** | **0.403** |
| Llama3 3B → Llama3 8B | Vanilla | 0.469 | 0.469 | 0.125 | 0.110 | 0.444 | 0.378 |
| | KD (5-shot) | 0.470 | **0.479** | 0.126 | 0.111 | 0.446 | 0.379 |
| | TITOK (ours) | **0.509** | 0.475 | **0.127** | **0.113** | **0.457** | **0.392** |
| Llama2 7B → Llama3 8B | Vanilla | 0.469 | 0.469 | 0.125 | 0.110 | 0.444 | 0.378 |
| | KD (5-shot) | 0.470 | 0.477 | 0.126 | 0.111 | 0.446 | 0.380 |
| | TITOK (ours) | **0.510** | **0.479** | **0.140** | **0.122** | **0.461** | **0.404** |

# E  COMPARISON WITH ADDITIONAL BASELINE

Here, we introduce an additional baseline that leverages few-shot supervision. In practice, our synthetic data generation process is seeded with a small set of prompts, which can be regarded as few-shot data. To be thorough, we therefore establish a baseline trained only on these five seed prompts, referred to as KD (5-shot), so that our evaluation of TITOK also considers minimal few-shot supervision. The results of this additional baseline are presented in Table 10.

Across all transfer settings, KD on only 5-shot samples provides only marginal improvements over the vanilla model, with average gains of 0.9% on reasoning tasks (BBH and MMLU) and 0.6–1.2% on personalization tasks (News Headline and Scholarly Title Generation). In contrast, TITOK delivers substantial improvements over this baseline, achieving 4.7% and 2.1% average gains on reasoning tasks and 8.9–16.6% average gains on personalization tasks. Winning over KD with only 5 few-shot samples shows that TITOK is not simply the result of a few-shot effect. Instead, the five seed prompts act only as a starting point, which our framework expands into richer and more effective synthetic training signals. This finding confirms that the true strength of TITOK lies in how it effectively leverages limited supervision rather than in the few-shot data itself.

# F  EXTENDED TABLE 1 RESULTS WITH VARIANCE AND STATISTICAL DETAIL

We additionally report the mean ± standard deviation over 3 random seeds of our results in Table 1. The results are presented in Table 11. As shown, the overall trend remains consistent: TITOK improves by 9.94% over the vanilla target model, 8.5% over KD, and 4.44% over TransLoRA. In fact, these improvements remain substantially larger than the corresponding standard deviations

# G  COMPREHENSIVE k% SENSITIVITY ANALYSIS

In this section, we conduct an extensive experiment on TITOK's performance throughout a wide range of k values, from 10% to 90%. The results are comprehensively presented in Table 12. These experiments demonstrate that TITOK consistently outperforms the target vanilla baseline throughout a broad, steady range of values, while the exact performance varies slightly depending on k%. This clearly indicates that the method is not overly sensitive to the choice of k% and that there exists a broad range of reasonable k% values where improvements are reliably obtained.

Table 11: **Main results with variance.** Mean $\pm$ standard deviation over three random seeds on BBH, MMLU, News Headline, and Scholarly Title under four transfer settings. BBH and MMLU are evaluated by exact-match accuracy, while News Headline and Scholarly Title are evaluated by ROUGE-1/L. All evaluations are zero-shot. Best scores are in **bold**, while second highest are underlined.

| Transfer | Method | BBH | MMLU | News Headline | | Scholarly Title | |
|---|---|---|---|---|---|---|---|
| | | Acc. | Acc. | R-1 | R-L | R-1 | R-L |
| Mistral 7B $\to$ Mistral 7B | Vanilla | 0.397 | 0.557 | 0.117 | 0.101 | 0.381 | 0.311 |
| | KD | 0.417 ± 0.007 | 0.560 ± 0.003 | 0.117 ± 0.004 | 0.104 ± 0.003 | 0.385 ± 0.006 | 0.310 ± 0.005 |
| | TransLoRA | 0.416 ± 0.006 | 0.534 ± 0.001 | 0.156 ± 0.002 | 0.137 ± 0.001 | 0.447 ± 0.001 | 0.382 ± 0.001 |
| | TɪTok (ours, k=70%) | **0.424** ± 0.008 | **0.561** ± 0.002 | **0.161** ± 0.001 | **0.143** ± 0.001 | **0.473** ± 0 | **0.413** ± 0.001 |
| Mistral 7B $\to$ Llama3 8B | Vanilla | 0.469 | 0.469 | 0.125 | 0.110 | 0.444 | 0.378 |
| | KD | 0.475 ± 0.004 | 0.482 ± 0.004 | 0.127 ± 0 | 0.112 ± 0.001 | 0.454 ± 0.001 | 0.387 ± 0.002 |
| | TransLoRA | 0.473 ± 0.003 | 0.473 ± 0.001 | 0.126 ± 0.001 | 0.110 ± 0.002 | 0.461 ± 0.001 | 0.397 ± 0.001 |
| | TɪTok (ours, k=70%) | **0.484** ± 0.002 | **0.485** ± 0.003 | **0.139** ± 0.001 | **0.123** ± 0.001 | **0.464** ± 0.001 | **0.403** ± 0.001 |
| Llama3 3B $\to$ Llama3 8B | Vanilla | 0.469 | 0.469 | 0.125 | 0.110 | 0.444 | 0.378 |
| | KD | 0.474 ± 0.005 | 0.477 ± 0.002 | 0.125 ± 0.001 | 0.110 ± 0.001 | 0.449 ± 0.001 | 0.383 ± 0.001 |
| | TransLoRA | 0.471 ± 0.009 | 0.467 ± 0.002 | 0.122 ± 0.001 | 0.108 ± 0.001 | 0.454 ± 0.001 | 0.387 ± 0.001 |
| | TɪTok (ours, k=30%) | **0.496** ± 0.011 | **0.478** ± 0.004 | **0.127** ± 0.001 | **0.113** ± 0.001 | **0.456** ± 0.001 | **0.392** ± 0 |
| Llama2 7B $\to$ Llama3 8B | Vanilla | 0.469 | 0.469 | 0.125 | 0.110 | 0.444 | 0.378 |
| | KD | 0.473 ± 0.002 | 0.476 ± 0.002 | 0.125 ± 0 | 0.110 ± 0.001 | 0.449 ± 0.001 | 0.382 ± 0.002 |
| | TransLoRA | 0.472 ± 0.002 | 0.468 ± 0.002 | 0.123 ± 0.001 | 0.109 ± 0.001 | 0.457 ± 0.003 | 0.394 ± 0.005 |
| | TɪTok (ours, k=70%) | **0.488** ± 0.019 | **0.477** ± 0.003 | **0.138** ± 0.002 | **0.120** ± 0.002 | **0.461** ± 0.001 | **0.403** ± 0.001 |

Table 12: TɪTok performance across $k = 10\%$–$90\%$. "Vanilla" denotes performance of target vanilla model. Highlighted columns denote the selected universal $k\%$ reported in Table 1. Highest performance are in **bold**, while the second best results are underlined.

| Transfer | Task | Metric | Vanilla | 10% | 20% | 30% | 40% | 50% | 60% | 70% | 80% | 90% |
|---|---|---|---|---|---|---|---|---|---|---|---|---|
| Mistral 7B $\to$ Mistral 7B | BBH | Acc | 0.397 | 0.401 | 0.420 | 0.444 | **0.445** | 0.441 | 0.432 | 0.432 | 0.431 | 0.431 |
| | MMLU | Acc | 0.557 | 0.556 | 0.556 | 0.558 | 0.553 | 0.554 | 0.560 | **0.563** | 0.560 | 0.558 |
| | News Headline | R-1 | 0.117 | 0.153 | 0.159 | **0.161** | 0.161 | 0.160 | 0.160 | 0.160 | 0.160 | 0.161 |
| | | R-L | 0.101 | 0.138 | 0.143 | **0.144** | 0.143 | 0.142 | 0.142 | 0.142 | 0.142 | 0.142 |
| | Scholarly Title | R-1 | 0.381 | 0.466 | 0.475 | **0.481** | 0.474 | 0.473 | 0.473 | 0.473 | 0.473 | 0.470 |
| | | R-L | 0.311 | 0.408 | 0.419 | **0.424** | 0.416 | 0.414 | 0.414 | 0.414 | 0.414 | 0.411 |
| Mistral 7B $\to$ Llama3 8B | BBH | Acc | 0.469 | 0.470 | 0.482 | 0.471 | 0.473 | 0.475 | 0.476 | 0.482 | **0.483** | 0.478 |
| | MMLU | Acc | 0.469 | 0.487 | 0.483 | **0.500** | 0.492 | 0.492 | 0.488 | 0.488 | 0.494 | **0.500** |
| | News Headline | R-1 | 0.125 | 0.140 | 0.141 | **0.142** | 0.142 | 0.141 | 0.140 | 0.140 | 0.138 | 0.138 |
| | | R-L | 0.110 | 0.123 | 0.124 | 0.125 | 0.126 | 0.124 | 0.123 | 0.124 | 0.122 | 0.121 |
| | Scholarly Title | R-1 | 0.444 | 0.460 | 0.458 | 0.458 | 0.460 | 0.467 | 0.466 | 0.464 | 0.465 | 0.465 |
| | | R-L | 0.378 | 0.398 | 0.394 | 0.395 | 0.396 | 0.406 | 0.405 | 0.403 | 0.403 | 0.406 |
| Llama3 3B $\to$ Llama3 8B | BBH | Acc | 0.469 | 0.515 | 0.507 | 0.509 | 0.512 | 0.500 | 0.492 | 0.505 | **0.518** | 0.509 |
| | MMLU | Acc | 0.469 | **0.479** | 0.477 | 0.475 | 0.475 | 0.475 | 0.474 | 0.475 | 0.474 | 0.472 |
| | News Headline | R-1 | 0.125 | 0.127 | 0.127 | 0.127 | **0.128** | 0.122 | 0.121 | 0.122 | 0.121 | 0.121 |
| | | R-L | 0.110 | 0.111 | 0.112 | **0.113** | 0.113 | 0.108 | 0.107 | 0.107 | 0.107 | 0.107 |
| | Scholarly Title | R-1 | 0.444 | 0.449 | 0.456 | 0.457 | 0.460 | **0.462** | 0.462 | 0.461 | **0.462** | **0.462** |
| | | R-L | 0.378 | 0.385 | 0.390 | 0.392 | 0.397 | **0.400** | 0.400 | 0.398 | 0.398 | 0.397 |
| Llama2 7B $\to$ Llama3 8B | BBH | Acc | 0.469 | 0.527 | 0.519 | **0.531** | 0.510 | 0.514 | 0.514 | 0.510 | 0.509 | 0.509 |
| | MMLU | Acc | 0.469 | 0.475 | 0.474 | 0.477 | 0.481 | 0.477 | **0.482** | 0.479 | 0.477 | 0.479 |
| | News Headline | R-1 | 0.125 | 0.126 | 0.129 | 0.130 | 0.135 | 0.138 | 0.138 | **0.140** | 0.139 | 0.139 |
| | | R-L | 0.110 | 0.110 | 0.114 | 0.115 | 0.119 | 0.121 | 0.120 | **0.122** | 0.121 | 0.121 |
| | Scholarly Title | R-1 | 0.444 | 0.450 | 0.455 | 0.453 | 0.453 | 0.458 | 0.460 | 0.461 | 0.461 | **0.464** |
| | | R-L | 0.378 | 0.385 | 0.391 | 0.389 | 0.389 | 0.396 | 0.402 | 0.404 | 0.404 | **0.406** |

The reason for choosing a universal k% hyperparameter in our study is to keep the presentation of our paper more coherent and consistent. For this reason, we intentionally avoid task-specific tuning and report the k% that generally works well across tasks. However, we note that with additional per-task optimization of k%, further performance improvements are indeed possible.

With regard to the adaptive mechanism, each transfer setting exhibits its own effective k% range. For instance, same-backbone BBH transfer favors a mid-range k% (30–70%), weak-to-strong BBH transfer benefits from smaller k%, and weak-to-strong News Headline Generation shows the oppo-

site trend, preferring larger k%. Taken together, these trends suggest that a simple adaptive mechanism can already be practical and effective. In our experiments, even coarse adjustments (i.e., using around 30% for weak-to-strong reasoning transfers and 70% for weak-to-strong stylistic tasks as denoted in Section 4.3) prove sufficient, without requiring extensive hyperparameter searches.

## H   SYNTHETIC DATA GENERATION WITH 2× POOL AND TOP-M SELECTION

In this section, we provide further details regarding the number of synthetic data samples used. In generating synthetic data, we first provide five samples from the original training set as few-shot exemplars to guide the model toward producing outputs in the desired style and format. For each task, we initially create a synthetic pool containing twice the number of examples used in the source model's training. This pool is then filtered using token-wise contrastive excess scores, after which we retain only the top $M$ samples, where $M$ equals the size of the source training set. To be specific, we set $M = 225$ for BBH, $M = 90$ for MMLU, and $M = 200$ for LaMP tasks. This procedure ensures that the target model's LoRA adapter is trained on a dataset comparable in scale to that of the source model, while selective filtering enhances the overall quality of the retained data.

## I   ROUGE-L FILTERING FOR DIVERSE SYNTHETIC QUERIES

We now proceed to provide the task lists for which we did not apply ROUGE-L filtering when generating diverse synthetic queries. In general, we use ROUGE-L filtering to encourage diversity in queries, but for the tasks listed in Table 13, we only applied simple deduplication. In the case of BBH, tasks such as *boolean_expressions* or *temporal_sequences* already follow highly restricted and repetitive patterns, making high ROUGE-L scores inevitable. For MMLU, the only tasks without ROUGE-L filtering are the history-related subjects. This is potentially because history questions often require long passages that overlap in vocabulary, phrasing, or factual references (*e.g.*, recurring names, dates, or events). Thus, applying strict ROUGE-L filtering in such cases made it difficult to generate the required number of synthetic queries. For all remaining BBH and MMLU tasks, as well as all LaMP tasks, we applied a ROUGE-L threshold of 0.7 to encourage diversity while still preserving task fidelity. Table 13 provides the tasks for which ROUGE-L filtering is not applied.

Table 13: Tasks without ROUGE-L filtering.

| Category | Tasks |
|---|---|
| BBH | boolean_expressions, date_understanding, disambiguation_qa, geometric_shapes, logical_deduction_three_objects, multistep_arithmetic_two, navigate, object_counting, penguins_in_a_table, reasoning_about_colored_objects, salient_translation_error_detection, snarks, temporal_sequences, tracking_shuffled_objects_three_objects, web_of_lies |
| MMLU | high_school_world_history, high_school_us_history, high_school_european_history |

## J   ROBUSTNESS OF TITOK TO SYNTHETIC DATA QUALITY

While TITOK's primary contribution is token-level transfer and is thus not tied to synthetic data, we provide a detailed analysis on the robustness of our method to synthetic data variations.

**Robustness to low-quality synthetic data**. First, we deliberately construct a suboptimal synthetic dataset by selecting only the 250 lowest-scoring synthetic samples, as evaluated by the `GPT-4o-mini` grading model and prompt in (scores $\in \{0, 1, 2, 3, 4, 5\}$) (Chen et al., 2024a). All training hyperparameters and filtering processes are identical to the original setup. As shown in Table 14, TITOK still substantially outperforms all baselines in the Mistral 7B → Mistral 7B BBH transfer setting. This shows that TITOK is still effective even with low-quality synthetic data.

| Model Transfer | Method | BBH (Acc.) |
|---|---|---|
| Mistral 7B → Mistral 7B | Vanilla | 0.397 |
| | KD | 0.406 |
| | TransLoRA | 0.405 |
| | TITOK (k=70%) | **0.416** |

Table 14: BBH exact-match performance for the Mistral 7B → Mistral 7B transfer setting using low-quality synthetic data. TITOK (k=70%) continues to outperform all baselines.

Table 15: Alternative synthetic query prompt template for BBH.

---
**Synthetic query generation prompt for {task_name} in BBH.**

---

Generate {task_name} questions like these examples:

Example 1:
(few-shot example 1)

Example 2:
(few-shot example 2)

Example 3:
(few-shot example 3)

Example 4:
(few-shot example 4)

Example 5:
(few-shot example 5)

Example 6:

---

**Robustness to data diversity**. Our synthetic data generation pipeline already incorporates deduplication and a ROUGE-L diversity threshold. This ensures the exclusion of exact duplicates and highly similar samples. More details are provided in Appendix I.

**Robustness to prompt design**. We further conduct an experiment with a different prompt for generating synthetic data. The alternative prompt template is provided in Table 15. We test this on BBH tasks in the Mistral 7B → Mistral 7B and Mistral 7B → Llama3 8B transfer settings.

Table 16 shows the results of using the alternative prompt for generating synthetic data. Analytically, the results show that TITOK achieves consistent improvements, and there are no substantial differences across prompt choices. This demonstrates that TITOK is not sensitive to prompt formulation, as long as the generated examples adhere to a reasonable and coherent structural pattern.

Overall, we emphasize that a highly refined or meticulously selected synthetic dataset is not required for TITOK. As long as the generated samples retain a relatively cohesive task structure, TITOK is still effective and is not very sensitive to the particular features of the synthetic data.

## K    ADDRESSING SYNTHETIC DATA QUALITY AND BIAS

We now move on to explain how our pipeline incorporates several components designed to reduce bias and prevent quality degradation in synthetic data. Specifically, we achieve this by incorporating several components, including a length filter that removes malformed or low-informative samples, a token-wise contrastive excess mechanism that selects only the most informative tokens rather than relying on the entire synthetic instances, and diversity filtering ( i.e., deduplication and ROUGE-L thresholding) to prevent mode collapse during synthetic data generation.

Table 16: Effect of prompt variation on TITOK on BBH in the Mistral 7B → Mistral 7B and Mistral 7B → Llama3 8B settings.

| Transfer | Method | Prompt | BBH (Acc.) |
|---|---|---|---|
| Mistral 7B → Mistral 7B | Vanilla | – | 0.397 |
| | TITOK (k=70%) | original (Table 22) | 0.432 |
| | | alternative (Table 15) | 0.436 |
| Mistral 7B → Llama3 8B | Vanilla | – | 0.469 |
| | TITOK (k=70%) | original (Table 22) | 0.482 |
| | | alternative(Table 15) | 0.485 |

| Data Source | BBH | Scholarly Title |
|---|---|---|
| Gold data | 2.40 | 3.68 |
| Mistral 7B synthetic data | 2.27 | 3.89 |
| Llama3 3B synthetic data | 2.01 | 3.74 |
| Llama2 7B_synthetic data | 2.07 | 3.15 |

Table 17: Overall average quality ratings of gold and synthetic datasets across the BBH and Scholarly Title Generation tasks. `GPT-4o-mini` evaluates each data sample on a 1–5 scale.

Additionally, we further execute an external quality evaluation using a separate model (`GPT-4o-mini`), adopting a robust synthetic data assessment methodology established in a prior work (Chen et al., 2024a). Table 17 reports the average quality scores (scale 1–5) of gold data and the synthetic data generated by multiple source models used in our experiments.

These results show that the synthetic data used in our experiments is comparable in quality to the gold dataset, with no evidence of harmful bias dominating the data. Importantly, TITOK performs consistently well across all synthetic sources, even when the quality varies (e.g., 2.01 vs. 2.40 in BBH), which further indicates that TITOK does not rely on perfectly clean or unbiased synthetic data. Taken together, these findings demonstrate that TITOK is robust to imperfections in synthetic data, and, once again, can be effectively used even in non-synthetic datasets as shown in Table 4.

## L  EXPLORATION ON SELF REFINEMENT

While TITOK primarily focuses on LoRA-to-LoRA knowledge transfer, we investigate its potential for self-refinement. We perform this experiment on BBH in the Mistral 7B → Mistral 7B and the Mistral 7B → Llama3 8B transfer settings. In particular, after the initial transfer step, 1) we generate synthetic data using the target LoRA that had already learned TITOK's transferred knowledge, 2) compute token-wise contrastive excess scores on the newly generated synthetic data using the original source LoRA and the its base model, and 3) train a new target LoRA with the same hyperparameters in the original setting (i.e., k=0.7). The results are summarized in Table 18.

Interestingly, the results show that the iterative step yields additional improvement, implying that iterative refining is possible. We hypothesize that this is possible because the initial transfer increases the target model's familiarity with the task and partially aligns its internal representations with the source domain. As a result, the target model generates cleaner, more coherent synthetic data with fewer irrelevant patterns. This thereby enables the second iteration of TITOK to more accurately identify domain-informative tokens, which improves token selection and makes excess scoring more discriminative. However, we emphasize that iterative refining synthetic data cannot surpass a target model trained on real train data. By creating increasingly better synthetic data, iteration can help narrow the gap, but it is still far from a fully supervised target model trained on actual data.

Table 18: "Initial TITOK" indicates TITOK applied once. "Iterative TITOK: *self transfer*" applies TITOK again using synthetic data generated by the learned target LoRA after the initial transfer.

| | | BBH (Acc.) |
|---|---|---|
| Mistral 7B → Mistral 7B | Vanilla | 0.397 |
| | Initial TITOK | 0.432 |
| | **Iterative TITOK: *self transfer*** | **0.456** |
| | Target expert | 0.460 |
| Mistral 7B → Llama3 8B | Vanilla | 0.469 |
| | Initial TITOK | 0.482 |
| | **Iterative TITOK: *self transfer*** | **0.510** |
| | Target expert | 0.531 |

| | | BBH | Scholarly_Title | |
|---|---|---|---|---|
| Model Transfer | Metric | Acc. | R-1 | R-L |
| | source lora | 0.460 | 0.429 | 0.363 |
| Llama3 3b → Llama3 8b | target base | 0.462 | 0.444 | 0.378 |
| | **TITOK** (k=30%) | **0.509** | **0.457** | **0.392** |
| | source lora | 0.359 | 0.431 | 0.367 |
| Llama2 7B → Llama3 8B | target base | 0.469 | 0.444 | 0.378 |
| | **TITOK** (k=70%) | **0.510** | **0.461** | **0.404** |

Table 19: Results on BBH and Scholarly Title Generation for transfer settings where the source LoRA is weaker than the target baseline.

## M    ROBUSTNESS OF TITOK UNDER WEAKER SOURCE LoRAS

We now discuss scenarios where the source LoRA is weaker than the target model. While it is true that TITOK uses the source LoRA as a reference, its effectiveness is not strongly dependent on the source LoRA's absolute performance. To illustrate this point more clearly, Table 19 summarizes the subsets of Table 1 in which the source LoRA underperforms the target model's vanilla version, and presents these source LoRA scores alongside the corresponding transfer results.

As shown in the table, TITOK still consistently improves the target model even when the source LoRA is weaker than the target baseline. This shows that TITOK captures domain-specific signals that are embedded in key tokens rather than inheriting the source LoRA's full behavior.

KD-based approaches, however, are highly dependent on the absolute capacity of the source LoRA. By making the target model follow the teacher's logits, a weak or poorly tuned source LoRA unavoidably transfers certain flaws, resulting in the minimal gains supported by the results in the above table. Meanwhile, TITOK avoids these deficiencies because it fundamentally differs in how it leverages the source LoRA. Instead of relying on the source LoRA's logits globally, TITOK uses the source only to smartly identify informative token positions. As a result, a poorly tuned source LoRA does not mislead TITOK since it does not make the target model imitate the source LoRA's behavior. Rather, it selectively retrieves the useful tokens using the source LoRA's domain knowledge.

## N    COMPUTATIONAL OVERHEAD COMPARISON

Here, we present the computational differences between TITOK and TransLoRA. We conduct an end-to-end time comparison of TransLoRA and TITOK on the BBH and News Headline Generation benchmarks in the Mistral 7B → Llama3 8B transfer setting. The results are presented in Table 20 and 21. Across both benchmarks, TITOK achieves roughly a 1.5×–2.5× reduction averagely in total compute time. Most of the gain comes from the removal of discriminator training, while the per-token log-likelihood computation produces only a minimal and manageable overhead.

**BBH (250 data × 27 tasks)**

| Method | Metric | discriminator training | selecting data | training | TOTAL |
|---|---|---|---|---|---|
| TransLoRA | avg_total (sec/num_task) | 279.33936 | 31.72915 | 104.61636 | 415.68487 |
| | avg_sample (sec/sample) | 0.63251 | 0.06346 | 0.41847 | 1.11444 |

| Method | Metric | token-wise contrastive excess score | selecting tokens (k 0.7) | token align | training | TOTAL |
|---|---|---|---|---|---|---|
| TɪToᴋ | avg_total (sec/num_task) | 164.70895 | 0.01155 | 0.41080 | 102.02867 | 267.15997 |
| | avg_sample (sec/sample) | 0.32942 | 0.00005 | 0.00054 | 0.40811 | 0.73812 |

Table 20: Runtime comparison for Mistral 7B → Llama3 8B transfer on BBH.

**News_Headline (200 data × 30 users)**

| Method | Metric | discriminator training | selecting data | training | TOTAL |
|---|---|---|---|---|---|
| TransLoRA | avg_total (sec/num_users) | 245.10085 | 25.39730 | 58.71148 | 329.20963 |
| | avg_sample (sec/sample) | 0.61279 | 0.06358 | 0.29356 | 0.96993 |

| Method | Metric | token-wise contrastive excess score | selecting tokens (k 0.7) | token align | training | TOTAL |
|---|---|---|---|---|---|---|
| TɪToᴋ | avg_total (sec/num_users) | 81.31437 | 0.01863 | 0.89304 | 49.52768 | 131.75371 |
| | avg_sample (sec/sample) | 0.20355 | 0.00005 | 0.00447 | 0.24764 | 0.45570 |

Table 21: Runtime comparison for Mistral 7B → Llama3 8B transfer on News Headline Generation.

## O  QUALITATIVE EXAMPLES

In this section, we provide qualitative examples from the News Headline Generation in the Mistral 7B → Mistral 7B transfer setting to illustrate how TɪToᴋ identifies the most informative tokens. The qualitative examples are presented in Figure 5. Analytically, the selected tokens correspond to the beginnings of major semantic or structural units in the headline. These tokens carry the highest information value: they denote transitions, introduce core noun phrase components, or contain root words. Moreover, since the token-wise contrastive excess score is defined as the difference between the expert model (source LoRA + base) and the amateur model (base only), it pinpoints the tokens where the expert and amateur diverge most in their predictions and so include effective learning signals for target tasks. Taken together, these qualitative examples effectively demonstrate that TɪToᴋ consistently retains the tokens that are most important for providing structure and meaning.

## P  PROMPTS FOR SYNTHETIC QUERY GENERATION

In this section, we present the prompts used for generating synthetic queries. Each prompt includes five examples, which correspond to the seed prompts taken from the original data. For BBH, which consists of multiple subtasks, we specify task-specific instructions and formatting rules so that the generated queries and labels are structured according to the expected format. For example, Table 22 illustrates the case of *boolean_expressions* as a representative example in detail.

In contrast, for MMLU, a single unified prompt format is sufficient for most subjects. The exceptions are the history-related tasks, namely, *high_school_us_history*, *high_school_world_history*, and *high_school_european_history*. For these tasks, it is observed that more specific prompting is required to obtain better generations. The general MMLU prompt is shown in Table 23, whereas the history-related tasks make use of the specialized prompt provided in Table 24.

Finally, for LaMP tasks, we apply the same query template uniformly to each user. The detailed prompt templates used for this benchmark are provided in Tables 25 and 26.

| Prompt | You are a news headline generator.

Generate a headline for the following article.

article: With a scalloped hem, it's a show-stopping piece that's both unique and affordable. Want more? Be sure to check out Stylelist on…. 

headline: |
|---|---|
| Output | Stylist**': The Perfect White** Dress **For** Your **Wed**ding **(PHOT**OS) **** |

| Prompt | You are a news headline generator.

Generate a headline for the following article.

article: It's hard to imagine that you would be interested in hearing about my favorite holiday tradition, but if you are, I'd be happy to share….. 

headline: |
|---|---|
| Output | **How To** Make Your **Own DI**Y **Christmas** Or**naments (PH**OTOS) **** |

Figure 5: **Qualitative examples from News Headline Generation in the Mistral 7B → Mistral 7B setting**. **Bold** and highlighted tokens are part of the selected tokens. Notably, these tokens exhibit high token-wise contrastive excess scores and thus fall into the top 70% selected for training.

Table 22: Synthetic query prompt for the BBH *boolean_expressions* task. Each BBH subtask requires task-specific instructions and formatting rules to ensure that generated queries and labels follow the expected format. We show *boolean_expressions* task here as a representative example.

---

**Synthetic query generation prompt for BBH (*boolean_expressions*) task.**

---

**System**
You are an expert task generator for boolean_expressions tasks. Generate boolean expression evaluation tasks ending with ` is` – SINGLE LINE ONLY.

**CRITICAL FORMAT REQUIREMENTS:**
- Follow the EXACT format structure shown in the examples.
- ABSOLUTELY CRITICAL: Generate **ONLY ONE LINE**, ONE TASK per response.
- NO multiple lines, NO newlines (\n), NO multiple expressions.
- Must end with ` is`.
- Example of **INVALID** output: `True or False is\n\n False and True is`
- Example of **VALID** output: `True or False or not False and ( True or False ) is`

Generate diverse content but maintain the exact same format structure. Only output the task input, not the solution.

**User**
Generate new `boolean_expressions` tasks following these exact format examples:
Example 1: [boolean expression ending with ` is`]
Example 2: [boolean expression ending with ` is`]
Example 3: [boolean expression ending with ` is`]
Example 4: [boolean expression ending with ` is`]
Example 5: [boolean expression ending with ` is`]

CRITICAL: Generate ONLY ONE LINE, exactly like the examples above.

Generate a new task following the exact format:

---

Table 23: Synthetic query prompt for the MMLU. Applied to all tasks except for history related tasks.

---

**Synthetic query generation prompt for MMLU (general).**

---

Example 1:
Question: [question snippet]
1. [choice snippet]
2. [choice snippet]
3. [choice snippet]
4. [choice snippet]
Answer: [1–4]

Example 2:
Question: [question snippet]
1. [choice snippet]
2. [choice snippet]
3. [choice snippet]
4. [choice snippet]
Answer: [1–4]

Example 3:
Question: [question snippet]
1. [choice snippet]
2. [choice snippet]
3. [choice snippet]
4. [choice snippet]
Answer: [1–4]

Example 4:
Question: [question snippet]
1. [choice snippet]
2. [choice snippet]
3. [choice snippet]
4. [choice snippet]
Answer: [1–4]

Example 5:
Question: [question snippet]
1. [choice snippet]
2. [choice snippet]
3. [choice snippet]
4. [choice snippet]
Answer: [1–4]

Example 6:

---

Table 24: Synthetic query prompt for the MMLU history tasks only. The history tasks are *high_school_us_history*, *high_school_world_history*, and *high_school_european_history*. These history tasks required more specific prompting to obtain better results.

---

**Synthetic query generation prompt for MMLU (history related tasks).**

---

Generate [SUBJECT] multiple choice questions following these examples:

Example 1:
Question: [history question snippet]
1. [choice snippet]
2. [choice snippet]
3. [choice snippet]
4. [choice snippet]
Answer: [1–4]

Example 2:
Question: [history question snippet]
1. [choice snippet]
2. [choice snippet]
3. [choice snippet]
4. [choice snippet]
Answer: [1–4]

Example 3:
Question: [history question snippet]
1. [choice snippet]
2. [choice snippet]
3. [choice snippet]
4. [choice snippet]
Answer: [1–4]

Example 4:
Question: [history question snippet]
1. [choice snippet]
2. [choice snippet]
3. [choice snippet]
4. [choice snippet]
Answer: [1–4]

Example 5:
Question: [history question snippet]
1. [choice snippet]
2. [choice snippet]
3. [choice snippet]
4. [choice snippet]
Answer: [1–4]

Now generate a new [SUBJECT] question following the same format:
Question:

---

Table 25: Synthetic query prompt for the News Headline Generation task.

| Synthetic query generation prompt for News Headline Generation |
| --- |
| You are a news text generator. Generate diverse news article texts that could be used to create headlines. Only output the raw news text content, not headlines or queries.

Example 1: [article snippet]
Example 2: [article snippet]
Example 3: [article snippet]
Example 4: [article snippet]
Example 5: [article snippet]

Example 6: |

Table 26: Synthetic query prompt for the Scholarly Title Generation task.

| Synthetic query generation prompt for Scholarly Title Generation |
| --- |
| You are a scholarly abstract generator. Generate diverse ONE paragraph abstracts that ask for creating paper titles from abstracts. The abstract should be ONE paragraph only. Only output the raw abstract text, not the actual titles.

Example 1: Create a title for this research paper:
Abstract: "[abstract snippet]"
Title: "[title]"

Example 2: Create a title for this research paper:
Abstract: "[abstract snippet]"
Title: "[title]"

Example 3: Create a title for this research paper:
Abstract: "[abstract snippet]"
Title: "[title]"

Example 4: Create a title for this research paper:
Abstract: "[abstract snippet]"
Title: "[title]"

Example 5: Create a title for this research paper:
Abstract: "[abstract snippet]"
Title: "[title]"

Write a research paper abstract as a single paragraph containing at least 3 sentences.

Example 6: |

