# OpenReview forum: "TiTok: Transfer Token-level Knowledge via Contrastive Excess to Transplant LoRA"
_ICLR.cc/2026/Conference — ICLR 2026 Poster_

### Official Review · Reviewer_g2oB · 2025-10-15

**Soundness:** 3
**Presentation:** 3
**Contribution:** 2
**Rating:** 4
**Confidence:** 3

**Summary:**

This paper introduces TITOK, a framework for transferring LoRA adapters between large language models through token-level contrastive knowledge transfer.
Unlike existing methods such as TransLoRA, which rely on synthetic data filtered by an external discriminator, TITOK uses a self-contained contrastive excess mechanism that compares token-level likelihoods between a source model with and without LoRA. Tokens where the LoRA contributes the most (“excessive” tokens) are identified as carrying rich, task-specific knowledge.

**Strengths:**

1. Lightweight and scalable.
TITOK requires no additional model training; it simply reuses the source model’s log-likelihoods. The proposed sample filtering and token selection can be implemented efficiently and applied to diverse architectures.

2. Cross-tokenizer alignment.
The tokenizer-alignment algorithm (Fig. 4) is an important practical contribution that allows reliable transfer across families like Mistral→LLaMA, addressing a frequent but rarely discussed issue in adapter transplantation.

3. Comprehensive evaluation.
The experiments cover both reasoning (BBH, MMLU) and personalization (LaMP) tasks, across intra- and inter-family transfers (same model, different family, size, and version). Ablations, sensitivity to token selection ratio, and external-data experiments are thorough and convincing.

4. Strong empirical results.
TITOK consistently improves over baselines (e.g., +7.11% average gain in cross-model transfer) and shows robustness even when transferring with unrelated data (Table 3).

**Weaknesses:**

1. Incremental relative to TransLoRA.
While the contrastive excess concept is novel, much of the overall framework (synthetic data, adapter transplantation) follows TransLoRA. The improvement may be viewed as a simplification rather than a fundamental rethinking of LoRA transfer.

2. Limited theoretical justification.
The paper could benefit from more formal analysis—e.g., why token-level contrastive loss correlates with adapter informativeness, or whether excess scores approximate information gain.

3. Evaluation scope.
All experiments use instruction-tuned LLMs up to 8B parameters. It remains unclear whether TITOK scales to larger frontier models (≥70B) or non-text modalities.

4. Minor clarity issues.
The term “contrastive excess” could be better linked to contrastive learning literature; at present it may be confusing since no contrastive pairs or negatives are explicitly optimized.

**Questions:**

See weaknesses

---

> ### Author Response · Authors · 2025-11-22
> **Response to Reviewer g2oB [1/3]**
>
> Dear Reviewer g2oB,
>
> We greatly appreciate your thoughtful and detailed analysis of our work. Below, we provide a thorough response to each of the points you raised. For clarity, we note that model names in our response are abbreviated as follows:
>
> * **Mistral 7B** = Mistral-7B-Instruct-v0.3, **Llama3 8B** = Llama-3.1-8B-Instruct, **Llama3 3B** = Llama-3.2-3B-Instruct, **Llama2 7B** = Llama-2-7b-chat-hf
>
>
> ---
>
> **[W1] Incremental relative to TransLoRA. While the contrastive excess concept is novel, much of the overall framework (synthetic data, adapter transplantation) follows TransLoRA. The improvement may be viewed as a simplification rather than a fundamental rethinking of LoRA transfer.**
>
> We understand why the shared use of synthetic data and the usage of adapter transplantation might make TiTok appear similar to TransLoRA [1] at first glance. However, we would like to respectfully clarify that the similarity is solely due to the common task environment, and TiTok's logic is fundamentally different from TransLoRA.
>
> * TransLoRA’s core assumption is to transfer entire synthetic sequences generated by the expert. Therefore, its framework is tightly coupled to synthetic data and is unable to function in the absence of teacher-generated labels. In that sense, this also implies that TransLoRA cannot be applied to external datasets. Thus, it focuses mainly on the role of synthetic data rather than the transfer mechanism itself.
>
> * In contrast, TiTok directly rethinks the mechanism of knowledge transfer itself. TiTok focuses on determining which tokens contain expert-specific knowledge (i.e  the tokens where the expert significantly deviates from base model) rather than copying entire synthetic sequences. Our novel  token-level contrastive signal fundamentally changes how knowledge is extracted and aligned. Because this process does not rely on teacher-generated labels, TiTok naturally extends to external, non-synthetic datasets (Table 3 in our paper) and is therefore more generalizable.
>
> Therefore, despite the fact that the objective of both approaches is to solve the same high-level LoRA transfer problem, which inevitably leads to certain overlapping components, the underlying philosophies are essentially different. TransLoRA focuses on transferring full outputs and relies entirely on synthetic data, whereas TiTok focuses on transferring the expert’s internal knowledge signals regardless of the data source. We believe this distinction is not a simplification but a meaningful conceptual advance, and we hope this clarification makes the intent and novelty of our contribution clear.
>
> We have added this explanation in Appendix D of the updated draft

---

> ### Author Response · Authors · 2025-11-22
> **Response to Reviewer g2oB [2/3]**
>
> **[W2] Limited theoretical justification. The paper could benefit from more formal analysis—e.g., why token-level contrastive loss correlates with adapter informativeness, or whether excess scores approximate information gain.**
>
> The proposed contrastive excess score is not a heuristic but a token-level log-likelihood ratio (LLR) between the source expert (backbone + LoRA) and the backbone. LLR is a robust metric widely established in statistical testing [2]; for example, by the Neyman–Pearson lemma [3], the likelihood ratio is known to be the optimal statistic for identifying differences between two models. Therefore, high-LLR tokens are exactly the regions where the adapter changes the predictive distribution, i.e., where task knowledge is injected; thus, selecting high-LLR tokens is theoretically justified as extracting the most informative adapter-specific signals.
>
> In addition, from the perspective of standard knowledge distillation, tokens with near-zero LLR provide no additional teacher knowledge and offer no benefit for transfer. In contrast, high-LLR tokens correspond to maximal teacher–student divergence, larger gradients, and higher information contribution to the student. Therefore, emphasizing tokens with high LLR naturally focuses training on the precise positions where transferable knowledge is concentrated.
>
> We verify this empirically by contrasting token subgroups of the top 20%, bottom 20%, and random 20% in the BBH Mistral 7B -> Llama3 8B setting. To prevent unintentional overlap across groups and to extract the most informative region, we employ a comparatively low 20% threshold. As it can be seen in the table below, the results of the experiment show that the top 20% produces the best transfer performance, thereby validating that tokens with high contrastive excess scores contain concentrated task knowledge.
>
>
> | Model Transfer        | Setting       | Variant       | Acc.   |
> |-----------------------|---------------|---------------|--------|
> | **Mistral 7B**   | Vanilla       |               | 0.469  |
> | → **Llama3 8B**                      | **TiTok** | **top 20%**    | **0.482** |
> |                       |               | bottom 20%    | 0.468  |
> |                       |               | random 20%    | 0.476  |
>
>
> Furthermore, we demonstrate that truly important tokens are consistently selected regardless of the expert's model.  For example, for BBH datasets, Mistral 7B and Llama2 7B experts agree on 59.76% of the chosen tokens even when we limit the selection to a small top 20% subset to prevent trivial overlap. This implies that a token will continue to be recognized across models if it is truly significant for the task, and it can be identified by our contrastive excess score.
>
> We add this result and the corresponding discussion in Appendix B of the revised draft.
>
>
> ----
>
> **[W3] Evaluation scope. All experiments use instruction-tuned LLMs up to 8B parameters. It remains unclear whether TITOK scales to larger frontier models (≥70B) or non-text modalities.**
>
> Following your suggestion, we conducted an experiment in the Llama-3.3-70B-Instruct -> Llama-3.3-70B-Instruct transfer setting. We conducted under 4-bit quantization due to the large model sizes involved.
>
>
> | Model Transfer | Metric  | Acc.  |
> |----------------|---------|-------|
> | `Llama-3.3-70B-Instruct` | Vanilla | 0.593 |
> | → `Llama-3.3-70B-Instruct` | **TiTok (70%)** | **0.621** |
>
> As it can be clearly seen in the table, TiTok effectively transfers knowledge even in the Llama-3.3-70B-Instruct -> Llama-3.3-70B-Instruct setting, with 4.7% improvement.This result shows that TiTok generalizes well to very large models, preserving its ability to transfer token-level knowledge effectively
>
> Regarding non-text modalities, this direction is beyond the scope of our current work, but we note that TiTok is conceptually compatible with any modality where the model is trained with token-level supervision. In such cases, the same token-level filtering principle could be applied, though we leave full multimodal validation for future work.
>
> We have included this result, along with the accompanying discussion, in Appendix J of the revised draft.

---

> ### Author Response · Authors · 2025-11-22
> **Response to Reviewer g2oB [3/3]**
>
> **[W4] Minor clarity issues. The term “contrastive excess” could be better linked to contrastive learning literature; at present it may be confusing since no contrastive pairs or negatives are explicitly optimized.**
>
> We thank the reviewer for constructive comments to prevent potential confusion. We agree that the term contrastive excess may suggest connections to contrastive learning, which has nothing to do with our work. To avoid confusion, we will rename the metric to **token-wise contrastive excess score**, given that our metric measures the token-wise likelihood excess of the expert model over the base model. We believe that adding the term "token-wise" will clarify the granularity of the signal and removes any unintended association with contrastive learning frameworks
>
> ---
>
> [1] Wang et al., Trans-LoRA: Towards Data-Free Transferable Parameter-Efficient Finetuning, NeurIPS 2024.
> [2] Li & Babu, A Graduate Course on Statistical Inference, Springer, 2019.
> [3] Neyman & Pearson, On the Most Efficient Tests of Statistical Hypotheses, Phil. Trans. R. Soc. A, 1933.
>
> ---
>
>
> We sincerely appreciate your thoughtful input and hope our responses provide the necessary clarity. If you have any further questions, please don't hesitate and let us know.
>
> Thank you very much,
> Authors

---

> > ### Comment · Reviewer_g2oB · 2025-11-23
> >
> > Thank the authors for their response. I will raise my score to weak accept.

---

> ### Author Response · Authors · 2025-11-24
> **Response to Reviewer g2oB**
>
> Dear Reviewer g2oB,
>
>
>
> Thank you for raising your score. We’re grateful that our explanation was helpful, and we truly appreciate your thoughtful feedback. Please don’t hesitate to ask if anything remains unclear.
>
>
> Sincerely,
> Authors

---

### Official Review · Reviewer_YLn5 · 2025-10-31

**Soundness:** 3
**Presentation:** 3
**Contribution:** 3
**Rating:** 6
**Confidence:** 4

**Summary:**

This paper proposes TITOK, a lightweight framework for transferring LoRA adapters across LLMs without original training data. It uses token-level contrastive excess scores from the difference between a base model and its LoRA version to filter synthetic data and focus training on informative tokens. TITOK avoids extra discriminators or full datasets, achieving consistent improvements over baselines like KD and TransLoRA across tasks and model types.

**Strengths:**

1. TITOK avoids extra models or full datasets by leveraging only token-level contrastive excess scores, making it computationally efficient and easy to deploy.

2. Demonstrates consistent performance gains across diverse transfer settings, including different model families, sizes, and tokenizers, highlighting robustness and broad applicability.

**Weaknesses:**

- Only 7–8 B models; no evidence it scales to 70 B+ or harder generative tasks.

- Source model generates both query and label; filter can’t catch wrong or biased synthetic data.

- No formal justification that “contrastive excess” equals transferable knowledge.

**Questions:**

Did the authors perform any human evaluation or external fact-checking to verify the quality of the filtered synthetic data?

---

> ### Author Response · Authors · 2025-11-22
> **Response to Reviewer YLn5 [1/2]**
>
> Dear Reviewer YLn5,
>
>
> We appreciate your insightful comments and the time dedicated to evaluating our work. We have addressed each point below.
>  For clarity, we note that model names in our response are abbreviated as follows:
>
> * **Mistral 7B** = Mistral-7B-Instruct-v0.3, **Llama3 8B** = Llama-3.1-8B-Instruct, **Llama3 3B** = Llama-3.2-3B-Instruct, **Llama2 7B** = Llama-2-7b-chat-hf
>
>
> ---
>
> **[W1] Only 7–8 B models; no evidence it scales to 70 B+ or harder generative tasks.**
>
> Following your suggestion, we conducted an experiment in the `Llama-3.3-70B-Instruct` → `Llama-3.3-70B-Instruct` transfer setting. We conducted under 4-bit quantization due to the large model sizes involved.
>
>
> | Model Transfer | Metric  | Acc.  |
> |----------------|---------|-------|
> | `Llama-3.3-70B-Instruct` | Vanilla | 0.593 |
> | → `Llama-3.3-70B-Instruct` | **TiTok (70%)** | **0.621** |
>
>
>
> As it can be clearly seen in the table, TiTok effectively transfers knowledge even in the `Llama-3.3-70B-Instruct` → `Llama-3.3-70B-Instruct` setting, with 4.7% improvement. This result shows that TiTok generalizes well to very large models, preserving its ability to transfer token-level knowledge effectively.
>
> We included this result and its discussion in Appendix J of the revised draft for completeness.
>
>
>
> ---
> **[W2, Q1] Source model generates both query and label; filter can’t catch wrong or biased synthetic data.**
>
> We would like to clarify that our primary contribution lies in token-level transfer, not in synthetic data generation. Our goal is to identify which tokens encode transferable knowledge and how these signals should be aligned across model families, rather than focusing mainly on utilizing synthetic data. In fact, as shown in Table 3 of our paper, TiTok continues to perform strongly even on external, non-synthetic datasets, demonstrating that our method is not tied to synthetic data. Moreover, TiTok remains effective even when the target model itself generates the synthetic queries (Figure 2), further confirming that its success does not depend on the quality or bias of a particular source generator.
>
> In addition, we would like to point out that our pipeline consists of several elements that are specifically made to reduce such issues:
>
> * a length filter to remove malformed or low-informative samples,
> * our token-wise contrastive excess mechanism filters and selects only the most informative tokens rather than relying on entire synthetic sentences, and
> * diversity filtering (e.g., deduplication and ROUGE-L thresholds) to avoid mode collapse in synthetic generation.
>
>
> Beyond these, to further address your concern regarding the quality of synthetic data,  we conducted an external quality evaluation using a separate model (GPT-4o-mini), adopting the robust synthetic-data assessment methodology established in prior work [1]. The table below reports the average quality scores (scale 1–5) of gold data and the synthetic data generated by multiple source models used in our experiments.
>
>
>
> | Dataset               |      | BBH  |        | Scholarly Title |
> |-------------------------|---|------:|-----------|------------:|
> | Gold data              |     | 2.40 |         | 3.68             |
> | Mistral 7B synthetic data  |    | 2.27   |         | 3.89             |
> | Llama3 3B synthetic data  |    | 2.01 |        |3.74             |
> | Llama2 7B synthetic data   |   | 2.07 |       |3.15             |
>
>
> These results show that the synthetic data used in our experiments is comparable in quality to the gold dataset, with no evidence of harmful bias dominating the data. Importantly, TiTok performs consistently well across all synthetic sources, even when the quality varies (e.g., 2.01 vs. 2.40 in BBH), which further indicates that TiTok does not rely on perfectly clean or unbiased synthetic data.
> Taken together, these findings demonstrate that TiTok is robust to imperfections in synthetic data, and, once again, can be effectively used even in non-synthetic datasets.
>
> We added the result and discussion to Appendix N in the updated draft.

---

> ### Author Response · Authors · 2025-11-22
> **Response to Reviewer YLn5 [2/2]**
>
> **[W3] No formal justification that “contrastive excess” equals transferable knowledge.**
>
> The proposed contrastive excess score is not a heuristic but a token-level log-likelihood ratio (LLR) between the source expert (backbone + LoRA) and the backbone. LLR is a robust metric widely established in statistical testing [2]; for example, by the Neyman–Pearson lemma [3], the likelihood ratio is known to be the optimal statistic for identifying differences between two models. Therefore, high-LLR tokens are exactly the regions where the adapter changes the predictive distribution, i.e., where task knowledge is injected; thus, selecting high-LLR tokens is theoretically justified as extracting the most informative adapter-specific signals.
>
> In addition, from the perspective of standard knowledge distillation, tokens with near-zero LLR provide no additional teacher knowledge and offer no benefit for transfer. In contrast, high-LLR tokens correspond to maximal teacher–student divergence, larger gradients, and higher information contribution to the student. Therefore, emphasizing tokens with high LLR naturally focuses training on the precise positions where transferable knowledge is concentrated.
>
> We verify this empirically by contrasting token subgroups of the top 20%, bottom 20%, and random 20% in the BBH Mistral 7B → Llama3 8B setting. To prevent unintentional overlap across groups and to extract the most informative region, we employ a comparatively low 20% threshold. As it can be seen in the table below, the results of the experiment show that the top 20% produces the best transfer performance, thereby validating that tokens with high contrastive excess scores contain concentrated task knowledge.
>
>
> | Model Transfer        | Setting       | Variant       | Acc.   |
> |-----------------------|---------------|---------------|--------|
> | **Mistral 7B**   | Vanilla       |               | 0.469  |
> | → **Llama3 8B**                      | **TiTok** | **top 20%**    | **0.482** |
> |                       |               | bottom 20%    | 0.468  |
> |                       |               | random 20%    | 0.476  |
>
>
> Furthermore, we demonstrate that truly important tokens are consistently selected regardless of the expert's model.  For example, for BBH datasets, Mistral 7B and Llama2 7B experts agree on 59.76% of the chosen tokens even when we limit the selection to a small top 20% subset to prevent trivial overlap. This implies that a token will continue to be recognized across models if it is truly significant for the task, and it can be identified by our contrastive excess score.
>
>
> The revised manuscript now includes this result and a detailed discussion in Appendix B.
>
>
> ---
>
> [1] Chen et al., AlpaGasus: Training a Better Alpaca with Fewer Data, ICLR 2024.
> [2] Li & Babu, A Graduate Course on Statistical Inference, Springer, 2019.
> [3] Neyman & Pearson, On the Most Efficient Tests of Statistical Hypotheses, Phil. Trans. R. Soc. A, 1933.
>
> ---
>
> We appreciate your insightful comments and hope our responses clarify the matters discussed. If you have any further questions, please don't hesitate and let us know.
>
>
>
>
> Thank you very much,
> Authors

---

> ### Comment · Reviewer_YLn5 · 2025-11-27
>
> Thanks for the response. But I do hope authors can take W&Q into the revision carefully. I will adjust my rating accordingly.

---

> > ### Author Response · Authors · 2025-11-27
> > **Response to Reviewer YLn5**
> >
> > Dear Reviewer YLn5,
> >
> >
> >
> > We deeply appreciate your insightful comments and the score raise. It means a lot to us that our clarifications were helpful. Your feedback is extremely valuable, and we will make sure to reflect it thoroughly in the camera-ready version. Should you have any additional questions or thoughts, please don't hesitate to let us know.
> >
> >
> > Sincerely,
> > Authors

---

### Official Review · Reviewer_LDW6 · 2025-10-31

**Soundness:** 2
**Presentation:** 2
**Contribution:** 2
**Rating:** 2
**Confidence:** 3

**Summary:**

This paper presents a framework called TITOK for transferring LoRA-based knowledge between large language models (LLMs). The authors note that existing parameter-efficient fine-tuning (PEFT) methods such as LoRA cannot be directly transplanted across different backbones, while knowledge distillation (KD) approaches depend heavily on access to original task data. TITOK introduces the concept of contrastive excess, computed between a source model with LoRA and its corresponding base model without LoRA, to identify task-informative tokens. Using these token-level signals, the method filters synthetic data and trains the target model’s new LoRA adapter only on the most informative tokens, avoiding the need for an additional discriminator or real training data. Experiments on BBH, MMLU, and LaMP benchmarks show that TITOK consistently outperforms KD and TransLoRA across multiple transfer settings, achieving average performance gains of 4–8%. Overall, the paper proposes a lightweight token-level transfer framework and demonstrates its effectiveness through empirical evaluation.

**Strengths:**

1. The paper introduces a new perspective on LoRA knowledge transfer by focusing on token-level contrastive signals (“contrastive excess”) rather than sequence-level alignment. This provides a fine-grained and model-agnostic approach that does not require additional networks or data annotations.
2. The authors conduct systematic experiments across multiple benchmarks (BBH, MMLU, and LaMP) and various transfer settings (intra-family, cross-family, and cross-version). The results consistently demonstrate that TITOK achieves stable improvements over established baselines such as KD and TransLoRA.

**Weaknesses:**

1. The paper does not provide a solid theoretical justification for why the proposed contrastive excess signal effectively captures transferable task knowledge. The formulation remains largely heuristic, and no analysis is presented to support its generalization properties or convergence behavior.

2. Although the experiments cover multiple datasets, the reported improvements (4–8%) are modest and sometimes within the margin of variance. The absence of standard deviations or statistical significance tests weakens the credibility of the claimed gains.

3. Despite claiming to reduce data dependence, TITOK still relies heavily on synthetic data generated from the source model. The paper does not adequately evaluate how data quality, diversity, or prompt design affect transfer performance, making reproducibility and robustness questionable.

4. The paper mainly compares against KD and TransLoRA but omits other recent or stronger PEFT transfer approaches. Without these comparisons, it is unclear whether TITOK’s benefits are meaningful beyond this narrow baseline set.

5. Several key implementation details—such as the choice of hyperparameters (e.g., k% token ratio) and the impact of tokenizer alignment—are only described qualitatively. The algorithmic intuition is under-explained, and the overall presentation lacks ablation depth, which limits interpretability and reproducibility.

**Questions:**

1. The paper lacks a clear explanation of why the contrastive excess between the LoRA-equipped and base models effectively captures transferable task knowledge. Could the authors provide theoretical reasoning or empirical evidence (e.g., gradient or information analysis) to support its validity?

2. TITOK heavily depends on synthetic data generated by the source model. How robust is the method to variations in data quality, prompt design, or model strength? Would TITOK still perform well if the source model or synthetic data are suboptimal?

3. The algorithm’s performance appears sensitive to the top-k% token selection and sample filtering. Can the authors provide quantitative sensitivity analyses or propose adaptive mechanisms to make TITOK more stable across tasks?

4. The tokenizer alignment process is crucial but not quantitatively evaluated. How accurate is this alignment across model families with different tokenization schemes? Could small alignment errors lead to degraded performance?

5. The comparisons only include KD and TransLoRA. To better position the contribution, could the authors include or discuss results against more recent PEFT transfer methods (e.g., KD-LoRA, AdapterFusion, DoRA)? This would clarify TITOK’s true advantage.

---

> ### Author Response · Authors · 2025-11-22
> **Response to Reviewer LDW6 [1/6]**
>
> Dear Reviewer LDW6,
>
> We are grateful for your careful and thorough review. We have examined each comment in depth and present our detailed responses below. For clarity, we note that model names in our response are abbreviated as follows:
>
> * **Mistral 7B** = Mistral-7B-Instruct-v0.3, **Llama3 8B** = Llama-3.1-8B-Instruct, **Llama3 3B** = Llama-3.2-3B-Instruct, **Llama2 7B** = Llama-2-7b-chat-hf
>
> ---
>
> **[W1, Q1] Explanation of why the contrastive excess effectively captures transferable task knowledge.**
>
> We appreciate the reviewer’s insightful comments. In fact, the proposed contrastive excess score is not a heuristic but a token-level log-likelihood ratio (LLR) between the source expert (backbone + LoRA) and the backbone. LLR is a robust metric widely established in statistical testing [1]; for example, by the Neyman–Pearson lemma [2], the likelihood ratio is known to be the optimal statistic for identifying differences between two models. Therefore, high-LLR tokens are exactly the regions where the adapter changes the predictive distribution, i.e., where task knowledge is injected; thus, selecting high-LLR tokens is theoretically justified as extracting the most informative adapter-specific signals.
>
> In addition, from the perspective of standard knowledge distillation, tokens with near-zero LLR provide no additional teacher knowledge and offer no benefit for transfer. In contrast, high-LLR tokens correspond to maximal teacher–student divergence, larger gradients, and higher information contribution to the student. Therefore, emphasizing tokens with high LLR naturally focuses training on the precise positions where transferable knowledge is concentrated.
>
> We verify this empirically by contrasting token subgroups of the top 20%, bottom 20%, and random 20% in the BBH Mistral 7B -> Llama3 8B setting. To prevent unintentional overlap across groups and to extract the most informative region, we employ a comparatively low 20% threshold. As it can be seen in the table below, the results of the experiment show that the top 20% produces the best transfer performance, thereby validating that tokens with high contrastive excess scores contain concentrated task knowledge.
>
> | Model Transfer        | Setting       | Variant       | Acc.   |
> |-----------------------|---------------|---------------|--------|
> | **Mistral 7B**   | Vanilla       |               | 0.469  |
> | → **Llama3 8B**                      | **TiTok** | **top 20%**    | **0.482** |
> |                       |               | bottom 20%    | 0.468  |
> |                       |               | random 20%    | 0.476  |
>
> Furthermore, we demonstrate that truly important tokens are consistently selected regardless of the expert's model. For example, for BBH datasets, Mistral 7B and Llama2 7B experts agree on 59.76% of the chosen tokens even when we limit the selection to a small top 20% subset to prevent trivial overlap. This implies that a token will continue to be recognized across models if it is truly significant for the task, and it can be identified by our contrastive excess score.
>
> We have included this analysis in Appendix B of the revised draft.

---

> ### Author Response · Authors · 2025-11-22
> **Response to Reviewer LDW6 [2/6]**
>
> **[W2] Reported improvements (4–8%) are modest and sometimes within the margin of variance. The absence of standard deviations or statistical significance tests weakens the credibility of the claimed gains.**
>
> First, we would like to clarify that, when computed using the scores reported in Table 1, KD [3,4] improves the target model by 2.01%, TransLoRA[5] by 4.76%, and TiTok by 10.52% across all transfer settings and benchmarks averagely. This shows that our method offers a significantly greater absolute improvement over current baselines.
>
> Nevertheless, to further address your concern and to simultaneously reinforce our experiment results, we additionally report the mean ± standard deviation over 3 random seeds in the table below. As shown, the overall trend remains consistent: relative to the target vanilla baseline, KD improves by 1.25%, TransLoRA by 5.28%, and TiTok by 9.94%. In fact, these improvements remain substantially larger than the corresponding standard deviations.
>
> Due to the limited space, we have abbreviated "News Headline" → "NH" and "Scholarly Title" → "ST".
>
> **Mistral 7B → Mistral 7B**
>
> | Method      | BBH Acc.     | MMLU Acc.    | NH R-1      | NH R-L      | ST R-1      | ST R-L      |
> |-------------|---------------|--------------|-------------|-------------|-------------|-------------|
> | Vanilla     | 0.397         | 0.557        | 0.117       | 0.101       | 0.381       | 0.311       |
> | KD          | 0.417 ± 0.007 | 0.560 ± 0.003| 0.117±0.004 | 0.104±0.003 | 0.385±0.006 | 0.310±0.005 |
> | TransLoRA   | 0.416 ± 0.006 | 0.534 ± 0.001| 0.156±0.002 | 0.137±0.001 | 0.447±0.001 | 0.382±0.001 |
> | **TiTok (k=70%)** | **0.424 ± 0.008** | **0.561 ± 0.002** | **0.161±0.001** | **0.143±0.001** | **0.473±0.000** | **0.413±0.001** |
>
>
> **Mistral 7B → Llama3 8B**
>
> | Method            | BBH Acc.        | MMLU Acc.       | NH R-1        | NH R-L        | ST R-1        | ST R-L        |
> |-------------------|------------------|------------------|----------------|----------------|----------------|----------------|
> | Vanilla           | 0.469            | 0.469            | 0.125          | 0.110          | 0.444          | 0.378          |
> | KD + MinED              | 0.475 ± 0.004    | 0.482 ± 0.004    | 0.127 ± 0.000  | 0.112 ± 0.001  | 0.454 ± 0.001  | 0.387 ± 0.002  |
> | TransLoRA         | 0.473 ± 0.003    | 0.473 ± 0.001    | 0.126 ± 0.003  | 0.110 ± 0.002  | 0.461 ± 0.001  | 0.397 ± 0.001  |
> | **TiTok (k=70%)** | **0.484 ± 0.002**| **0.485 ± 0.003**| **0.139 ± 0.001**| **0.123 ± 0.001**| **0.464 ± 0.001**| **0.403 ± 0.001** |
>
>
>
> **Llama3 3B → Llama3 8B**
>
> | Method            | BBH Acc.        | MMLU Acc.       | NH R-1        | NH R-L        | ST R-1        | ST R-L        |
> |-------------------|------------------|------------------|----------------|----------------|----------------|----------------|
> | Vanilla           | 0.469            | 0.469            | 0.125          | 0.110          | 0.444          | 0.378          |
> | KD                | 0.474 ± 0.005    | 0.477 ± 0.002    | 0.125 ± 0.001  | 0.110 ± 0.001  | 0.449 ± 0.001  | 0.383 ± 0.001  |
> | TransLoRA         | 0.471 ± 0.009    | 0.467 ± 0.002    | 0.122 ± 0.001  | 0.108 ± 0.001  | 0.454 ± 0.001  | 0.387 ± 0.001  |
> | **TiTok (k=30%)** | **0.496 ± 0.011**| **0.478 ± 0.004**| **0.127 ± 0.001**| **0.113 ± 0.001**| **0.456 ± 0.001**| **0.392 ± 0.000** |
>
>
>
> **Llama2 7B → Llama3 8B**
>
>
>
> | Method            | BBH Acc.        | MMLU Acc.       | NH R-1        | NH R-L        | ST R-1        | ST R-L        |
> |-------------------|------------------|------------------|----------------|----------------|----------------|----------------|
> | Vanilla           | 0.469            | 0.469            | 0.125          | 0.110          | 0.444          | 0.378          |
> | KD  + MinED              | 0.473 ± 0.002    | 0.476 ± 0.002    | 0.125 ± 0.000  | 0.110 ± 0.001  | 0.449 ± 0.001  | 0.382 ± 0.002  |
> | TransLoRA         | 0.472 ± 0.002    | 0.468 ± 0.002    | 0.123 ± 0.001  | 0.109 ± 0.001  | 0.457 ± 0.003  | 0.394 ± 0.005  |
> | **TiTok (k=70%)** | **0.488 ± 0.019**| **0.477 ± 0.003**| **0.138 ± 0.002**| **0.120 ± 0.002**| **0.461 ± 0.001**| **0.403 ± 0.001** |
>
> We will include these full results with standard deviations in our camera-ready version. Thank you for the suggestion.
>
> This result and its detailed discussion have been added to Appendix F in the revised version of the manuscript.

---

> ### Author Response · Authors · 2025-11-22
> **Response to Reviewer LDW6 [3/6]**
>
> **[W3, Q2] TITOK heavily depends on synthetic data generated by the source model. How robust is the method to variations in data quality, prompt design, or model strength? Would TITOK still perform well if the source model or synthetic data are suboptimal?**
>
> ***1. TITOK does not depend on synthetic data***
>
> First, we would like to clarify that the primary contribution of our approach is token-level transfer, not synthetic data. With a particular focus on which tokens should be transferred and how those signals should be aligned, our work attempts to address the question of how the knowledge-transfer process itself should be designed.  In fact, Table 3 of our paper contains experiments using external, non-synthetic datasets where the approach continues to perform strongly, further demonstrating that TiTok is not tied to synthetic data. We hope this clarifies that our method is neither limited nor dependent on synthetic data.
>
>
> ***2. TITOK is robust to variations***
>
> Next, to directly address your question regarding robustness to synthetic-data variations, we conducted several additional experiments specifically for these concerns.
>
> **(1) Robustness to low-quality synthetic data**: First, we deliberately constructed a suboptimal synthetic dataset by selecting only the 250 lowest-scoring synthetic samples, as evaluated by the GPT-4o-mini grading model and prompt in [6] (scores ∈ {0,1,2,3,4,5}). All training hyperparameters filtering processes for each method follow the original paper. As shown in the results below, TiTok still substantially outperforms all baselines in the Mistral 7B→Mistral 7B BBH transfer setting. This demonstrates that TiTok is still effective even with low-quality synthetic data. Additionally, we would like to once again emphasize that we can even use external, non-synthetic datasets as presented in Table 3 of our paper.
>
>
> | Model Transfer             | Method            | BBH (Acc.) |
> |---------------------------|-------------------|------------|
> | **Mistral 7B**  | Vanilla           | 0.397      |
> |  → **Mistral 7B**                         | KD                | 0.406      |
> |                           | TransLoRA         | 0.405      |
> |                           | **TiTok (k=70%)** | **0.416**  |
>
>
> **(2) Robustness to data diversity**: Our synthetic data generation pipeline already incorporates deduplication and a ROUGE-L diversity threshold. This ensures the exclusion of exact duplicates and highly similar samples.
>
> **(3) Robustness to prompt design**: We further conducted an experiment with a different prompt for generating synthetic data. The prompt template is shown below:
>
> """
>
>
> Generate {task_name} questions like these examples:
>
> Example 1:
> (few-shot example 1)
>
> Example 2:
> (few-shot example 2)
>
> Example 3:
> (few-shot example 3)
>
> Example 4:
> (few-shot example 4)
>
> Example 5:
> (few-shot example 5)
>
> Example 6:
>
>
> """
>
>
>
> | Transfer                   | Method            | Prompt        | BBH (Acc.) |
> |---------------------------|-------------------|---------------|------------|
> | **Mistral 7B**  | Vanilla           | –             | 0.397      |
> |  → **Mistral 7B**                         | **TiTok (k=70%)** | original      | 0.432      |
> |                           |                   | alternative   | 0.436      |
> |                           |                   |               |            |
> | **Mistral 7B** | Vanilla           | –             | 0.469      |
> |  →  **Llama3 8B**                         | **TiTok (k=70%)** | original      | 0.482      |
> |                           |                   | alternative   | 0.485      |
> |                           |                   |               |            |
>
>
>
> The results show that TiTok again achieves consistent improvements, and there are no substantial differences across prompt choices. This demonstrates that TiTok is not sensitive to prompt formulation, as long as the generated examples follow a reasonable structure.
>
>
> Overall, we emphasize that a highly refined or meticulously selected synthetic data is not required for TiTok. This means that as long as the generated samples retain a relatively cohesive task structure, the method is still effective and is not very sensitive to the particular features of the synthetic data. This is supported by our additional experiments presented above.
>
> We incorporated these analyses and results and the relevant analysis into Appendix M of the revised manuscript.

---

> ### Author Response · Authors · 2025-11-22
> **Response to Reviewer LDW6 [4/6]**
>
> **[W3, Q2] TITOK heavily depends on synthetic data generated by the source model. How robust is the method to variations in data quality, prompt design, or model strength? Would TITOK still perform well if the source model or synthetic data are suboptimal? (con't)**
>
> ***3. TITOK is robust to model strength***
>
> With regard to the case of having a suboptimal expert model, while it is true that TiTok uses the source LoRA as a reference, its effectiveness is not strongly dependent on the source LoRA’s absolute performance. Below, we additionally report the cases in Table 1 where the source LoRA underperforms the target model’s vanilla version, together with the source LoRA’s own scores. As shown in the table, TiTok still consistently improves the target model even when the source LoRA is weaker than the target baseline. This indicates that TiTok captures domain-specific signals that are embedded in certain tokens rather than extracting the entire capabilities of the source LoRA.
>
>
> | Model Transfer             |              Metric|  BBH        | Scholarly | Title |
> |---------------------------|--------------|------------:|----------:|-------:|
> |                           |              | Acc.       |     R-1   |   R-L |
> | **Llama3 3B**             | source lora  | 0.460      |     0.429 | 0.363 |
> | → **Llama3 8B**           | target base  | 0.462      |     0.444 | 0.378 |
> |                           | **TiTok (k=30%)** | **0.509** | **0.457** | **0.392** |
> |                           |              |            |           |       |
> | **Llama2 7B**             | source lora  | 0.359      |     0.431 | 0.367 |
> | → **Llama3 8B**           | target base  | 0.469      |     0.444 | 0.378 |
> |                           | **TiTok (k=70%)** | **0.510** | **0.461** | **0.404** |
>
> KD-based approaches, however, are highly dependent on the absolute capacity of the source LoRA. By making the target model follow the teacher's logits, a weak or poorly tuned source LoRA unavoidably transfers certain flaws, resulting in the minimal gains supported by the data in the table. Meanwhile, TiTok avoids these deficiencies because it fundamentally differs in how it leverages the source LoRA. Instead of relying on the source LoRA’s logits globally, TiTok uses the source only to smartly identify informative token positions. As a result, a poorly tuned source LoRA does not mislead TiTok since it does not make the target model imitate the source LoRA’s behavior. Rather, it selectively retrieves the useful tokens using the source LoRA's domain knowledge.
>
> We added the result and discussion to Appendix O in the updated draft.
>
> ---
>
> **[W4, Q5] Could the authors include or discuss results against more recent PEFT transfer methods (e.g., KD-LoRA, AdapterFusion, DoRA)?**
>
> We appreciate the suggestion to compare against such baselines, and our analyses are as follows.
>
> * **KD-LoRA** [7]: KD-LoRA can only operate when the source and target models share the same architecture family. Thus, the method fails to run in cross-family settings such as Mistral → Llama, which are a central focus of our paper. Nevertheless, we still include a comparison in the Mistral 7B → Mistral 7B BBH setting, where TiTok still outperforms KD-LoRA.
>
>
> | Transfer         |       | Method            | BBH (Acc.) |
> |-------------------------|----|--------------|--------------------|
> | **Mistral 7B**   |    |vanilla           | 0.397              |
> |  → **Mistral 7B**       |                | KD-LoRA       | 0.429              |
> |        |      | **TiTok (k=70%)**   |**0.432**          |
>
>
> * **AdapterFusion** [8]: We would like to clarify that AdapterFusion is designed for combining multiple source adapters, rather than for performing PEFT-based knowledge transfer. Because our setting involves transferring knowledge from a single source LoRA, AdapterFusion is fundamentally incompatible with our problem formulation.
>
>
>
> * **DoRA** [9]: DoRA is a variant of LoRA that improves adapter efficiency. It decomposes pretrained weights into magnitude and direction and updates only the direction via normalized low-rank adaptation. But, it is not a transfer algorithm and does not perform or facilitate LoRA-to-LoRA knowledge transfer.
>
> For these reasons, we respectfully argue that the suggested methods are hard to compare with TiTok.
>
> More broadly, the majority of PEFT transfer techniques assume that the source and target models are part of the same model family and architecture, and they typically do not function for cross-family transfer. Given that many of our experiments involve transferring between different model families, architectures, and sizes, direct comparison with these methods is not feasible. In addition, because our method assumes no access to real training data, identifying appropriate baselines becomes even more challenging. For these reasons, we believe TiTok fills an important gap and offers a new, complementary approach.

---

> ### Author Response · Authors · 2025-11-22
> **Response to Reviewer LDW6 [5/6]**
>
> **[W5, Q3] The algorithm’s performance appears sensitive to the top-k% token selection and sample filtering. Can the authors provide quantitative sensitivity analyses or propose adaptive mechanisms to make TITOK more stable across tasks?**
>
> With regard to sample filtering and token selection, we have already provided results and their effects in the ablation table (Table 2) and Figure 3 of our paper. In particular, we note that certain trends emerge regarding the optimal k% range depending on the transfer setting. However, to address your concern fully, we conducted an extensive experiment on TiTok's performance throughout a wide range of k values, from 10% to 90%. The best scores  are highlighted in **bold**.
>
> ### **Mistral 7B → Mistral 7B**
>
> | Task            | Metric | Vanilla | 10% | 20% | 30% | 40% | 50% | 60% | 70% | 80% | 90% |
> |-----------------|--------|---------|------|------|------|------|------|------|------|------|------|
> | BBH             | Acc    | 0.397 | 0.401 | 0.420 |  0.444  | **0.445** | 0.441 | 0.432 | 0.432 | 0.431 | 0.431 |
> | MMLU            | Acc    | 0.557 | 0.556 | 0.556 | 0.558 | 0.553 | 0.554 | 0.560  | **0.563** | 0.560  | 0.558 |
> | News Headline   | R-1    | 0.117 | 0.153 | 0.159 | **0.161** | **0.161** | 0.160 | 0.160 | 0.160 | 0.160 | **0.161** |
> |                 | R-L    | 0.101 | 0.138 | 0.143 | **0.144** | 0.143 | 0.142 | 0.142 | 0.142 | 0.142 | 0.142 |
> | Scholarly Title | R-1    | 0.381 | 0.466 | 0.475 | **0.481** | 0.474 | 0.473 | 0.473 | 0.473 | 0.473 | 0.470 |
> |                 | R-L    | 0.311 | 0.408 | 0.419 | **0.424** | 0.416 | 0.414 | 0.414 | 0.414 | 0.414 | 0.411 |
>
>
> ### **Mistral 7B → Llama3 8B**
>
> | Task            | Metric | Vanilla | 10% | 20% | 30% | 40% | 50% | 60% | 70% | 80% | 90% |
> |-----------------|--------|---------|------|------|------|------|------|------|------|------|------|
> | BBH             | Acc    | 0.469 | 0.470 | 0.482 | 0.471 | 0.473 | 0.475 | 0.476 | 0.482 | **0.483** | 0.478 |
> | MMLU            | Acc    | 0.469 | 0.487 | 0.483 | **0.500** | 0.492 | 0.492 | 0.488 | 0.488 | 0.494 | **0.500** |
> | News Headline   | R-1    | 0.125 | 0.140 | 0.141 | **0.142** | **0.142** | 0.141 | 0.140 | 0.140 | 0.138 | 0.138 |
> |                 | R-L    | 0.110 | 0.123 | 0.124 | 0.125 | **0.126** | 0.124 | 0.123 | 0.124 | 0.122 | 0.121 |
> | Scholarly Title | R-1    | 0.444 | 0.460 | 0.458 | 0.458 | 0.460 | **0.467** | 0.466 | 0.464 | 0.465 | 0.465 |
> |                 | R-L    | 0.378 | 0.398 | 0.394 | 0.395 | 0.396 | **0.406** | 0.405 | 0.403 | 0.403 | **0.406** |
>
>
> ### **Llama3 3B → Llama3 8B**
>
> | Task            | Metric | Vanilla | 10% | 20% | 30% | 40% | 50% | 60% | 70% | 80% | 90% |
> |-----------------|--------|---------|------|------|------|------|------|------|------|------|------|
> | BBH             | Acc    | 0.469 | 0.515 | 0.507 | 0.509 | 0.512 | 0.500 | 0.492 | 0.505 | **0.518** | 0.509 |
> | MMLU            | Acc    | 0.469 | **0.479** | 0.477 | 0.475 | 0.475 | 0.475 | 0.474 | 0.475 | 0.474 | 0.472 |
> | News Headline   | R-1    | 0.125 | 0.127 | 0.127 | 0.127 | **0.128** | 0.122 | 0.121 | 0.122 | 0.121 | 0.121 |
> |                 | R-L    | 0.110 | 0.111 | 0.112 | **0.113** | **0.113** | 0.108 | 0.107 | 0.107 | 0.107 | 0.107 |
> | Scholarly Title | R-1    | 0.444 | 0.449 | 0.456 | 0.457 | 0.460 | **0.462** | **0.462** | 0.461 | **0.462** | **0.462** |
> |                 | R-L    | 0.378 | 0.385 | 0.390 | 0.392 | 0.397 | **0.400** | **0.400** | 0.398 | 0.398 | 0.397 |
>
> ### **Llama2 7B → Llama3 8B**
>
> | Task            | Metric | Vanilla | 10% | 20% | 30% | 40% | 50% | 60% | 70% | 80% | 90% |
> |-----------------|--------|---------|------|------|------|------|------|------|------|------|------|
> | BBH             | Acc    | 0.469 | 0.527 | 0.519 | **0.531** | 0.510 | 0.514 | 0.514 | 0.510 | 0.509 | 0.509 |
> | MMLU            | Acc    | 0.469 | 0.475 | 0.474 | 0.477 | 0.481 | 0.477 | **0.482** | 0.479 | 0.477 | 0.479 |
> | News Headline   | R-1    | 0.125 | 0.126 | 0.129 | 0.130 | 0.135 | 0.138 | 0.138 | **0.140** | 0.139 | 0.139 |
> |                 | R-L    | 0.110 | 0.110 | 0.114 | 0.115 | 0.119 | 0.121 | 0.120 | **0.122** | 0.121 | 0.121 |
> | Scholarly Title | R-1    | 0.444 | 0.450 | 0.455 | 0.453 | 0.453 | 0.458 | 0.460 | 0.461 | 0.461 | **0.464** |
> |                 | R-L    | 0.378 | 0.385 | 0.391 | 0.389 | 0.389 | 0.396 | 0.402 | 0.404 | 0.404 | **0.406** |
>
> These experiments demonstrate that TiTok consistently outperforms the target vanilla baseline throughout a broad, steady range of values, while the exact performance varies slightly depending on k%. This indicates that the method is not overly sensitive to the choice of k% and that there exists a broad range of reasonable k% values where improvements are reliably obtained.

---

> ### Author Response · Authors · 2025-11-22
> **Response to Reviewer LDW6 [6/6]**
>
> **[W5, Q3] The algorithm’s performance appears sensitive to the top-k% token selection and sample filtering. Can the authors provide quantitative sensitivity analyses or propose adaptive mechanisms to make TITOK more stable across tasks? (con't)**
>
> The reason for choosing a universal k% hyperparameter in our study was to keep the presentation of our paper more coherent and consistent. For this reason, we intentionally avoided task-specific tuning and reported the k% that generally works well across tasks. However, we note that additional per-task optimization of k% could further improve performance. We will add this table in our camera-ready version. Thank you for pointing this out.
>
> With regard to the adaptive mechanism, each transfer setting exhibits its own effective k% range. For instance, as explained in the paper, same-backbone BBH transfer favors a mid-range k% (40–70%), weak-to-strong BBH transfer benefits from smaller k%, and weak-to-strong News Headline Generation shows the opposite trend, preferring larger k%. Taken together, these trends suggest that a simple adaptive mechanism can already be practical and effective. In our experiments, even coarse adjustments (i.e  using around 30% for weak-to-strong reasoning transfers  and around 70% for weak-to-strong stylistic tasks as denoted in lines 435~445) proved sufficient, without requiring extensive hyperparameter searches.
>
> The revised manuscript now includes this result and a detailed discussion in Appendix H.
>
> ---
>
> **[W5, Q4] The tokenizer alignment process is crucial but not quantitatively evaluated. How accurate is this alignment across model families with different tokenization schemes? Could small alignment errors lead to degraded performance?**
>
>
> We would like to clarify that our tokenizer alignment algorithm is conceptually error-free because it simply performs a deterministic mapping between two tokenizers on the exact same text sequence. Since both tokenizations correspond to the identical underlying character string, the mapping cannot introduce semantic errors; every token in both tokenizers is defined over non-overlapping spans of the same text, and these spans align uniquely. For instance, in the Mistral 7B -> Llama3 8B transfer setting on BBH and news headline benchmarks, 100% of tokens are aligned, and there were **NO** exception cases as shown in the table below, confirming that alignment errors do not occur in practice.
>
>
> | Alignment Case | BBH    | News Headline |
> |----------------|--------|----------------|
> | exceptions     | 0%     | 0%             |
> | many to many   | 46.18% | 5.00%          |
> | many to one    | 3.47%  | 17.45%         |
> | one to many    | 0.01%  | 0.01%          |
> | one to one     | 50.33% | 77.52%         |
>
>
> To further evaluate robustness, we additionally constructed a degraded setting in which we keep only the one-to-one aligned token pairs and discard all other alignment cases. Due to limited spaces, we abbreviated "News Headline" to "NH".
>
>
> | Transfer                     | Method            | BBH Acc. | NH R-1 | NH R-L |
> |-----------------------------|-------------------|----------|--------|--------|
> | **Mistral-7B** | Vanilla           | 0.397    | 0.117  | 0.101  |
> |    → **Llama-3 8B**       | One to one only   | 0.472    | 0.138  | 0.120  |
> |                             | **TiTok (k=70%)** | **0.482**| **0.160** | **0.142** |
>
>
> ***​***
>
> The experiment result shows that using all the full alignment cases outperforms using one-to-one token pairs only. This indicates that the many-to-one, one-to-many, and many-to-many alignments are also correctly aligned, as including them yields the best results. Together, these results confirm that our alignment method is both accurate and robust.
>
> We have updated the manuscript by adding this result and the relevant explanation in Appendix A.
>
>
> ---
>
> [1] Li & Babu. A Graduate Course on Statistical Inference. Springer, 2019.
> [2] Neyman & Pearson. On the Problem of the Most Efficient Tests of Statistical Hypotheses. Phil. Trans. Royal Soc. A, 1933.
> [3] Wan et al., Knowledge Fusion of Large Language Models, ICLR 2024.
> [4] Hinton et al., Distilling the Knowledge in a Neural Network, NeurIPS 2014 Workshop.
> [5] Wang et al., Trans-LoRA: Towards Data-Free Transferable Parameter-Efficient Finetuning, NeurIPS 2024.
> [6] Chen et al., AlpaGasus: Training a Better Alpaca with Fewer Data, ICLR 2024.
> [7] Azimi et al., KD-LoRA: A Hybrid Approach to Efficient Fine-Tuning with LoRA and Knowledge Distillation, NeurIPS 2024 (ENLSP-IV Workshop).
> [8] Pfeiffer et al., AdapterFusion: Non-Destructive Task Composition for Transfer Learning, EACL 2021.
> [9] Liu et al., DoRA: Weight-Decomposed Low-Rank Adaptation, ICML 2024 (Oral).
>
> ---
>
>
> We hope our responses resolve the points you raised. If you have any further questions, please don't hesitate and let us know.
>
>
>
> Thank you very much,
> Authors

---

> > ### Comment · Reviewer_LDW6 · 2025-11-28
> >
> > Thank you for your response, which resolves my issue. I raise the score to “Accept.”

---

> ### Author Response · Authors · 2025-11-29
> **Response to Reviewer LDW6**
>
> Dear Reviewer LDW6,
>
>
> Thank you so much for your reconsideration and your score raise. We’re grateful not only for the updated score, but also for the time and attention you dedicated to understand our explanation. If any additional questions arise later on, please know we’re always here to help.
>
>
> Sincerely,
> Authors

---

### Official Review · Reviewer_AmyZ · 2025-11-01

**Soundness:** 3
**Presentation:** 3
**Contribution:** 3
**Rating:** 6
**Confidence:** 2

**Summary:**

This paper introduces TITOK (Transfer Token-level Knowledge), a framework for transplanting LoRA adapters across different LLM backbones by transferring token-level knowledge instead of sequence-level knowledge.
Unlike previous approaches such as TransLoRA, which require a discriminator model to filter synthetic data, TITOK leverages a contrastive excess signal to identify informative tokens that encode task-specific knowledge. TITOK requires neither extra models nor additional training overhead. It comes with an effective mechanism to resolve tokenizer mismatches between source and target models, which enhances robustness and applicability.

**Strengths:**

- The idea of contrastive excess as a proxy for task-specific token importance is elegant and avoids the need for auxiliary discriminators or extra training modules. It’s conceptually sound and efficiently leverages existing model behavior.
- TITOK uses no extra models and operates with modest computational overhead compared to TransLoRA. This design aligns well with real-world deployment constraints.
- The authors test across multiple model families (Mistral, Llama 2/3) and tasks (reasoning, personalization), presenting clear gains in each. The inclusion of ablations and sensitivity analyses (on token selection ratio, query generation source, etc.) adds strong empirical credibility.
- The dual-pointer alignment and averaging rules are simple yet effective.

**Weaknesses:**

- The proposed method still relies on generating a large pool of synthetic samples, which may introduce significant costs
- While token-level filtering is the paper’s key innovation, there is no qualitative or visual analysis demonstrating which tokens are retained or why they contribute to effective transfer.

**Questions:**

- How sensitive is TITOK to the quality of the source LoRA? Would a poorly tuned LoRA produce misleading excess scores?
- Could contrastive excess be used iteratively (e.g., to refine the LoRA itself) rather than just for one-shot transfer?
- How does TITOK behave when transferring between models with different architectures (e.g., dense vs. mixture-of-experts backbones)?
- Can the authors clarify the computational overhead of computing per-token log-likelihood differences compared to TransLoRA’s discriminator?

---

> ### Author Response · Authors · 2025-11-22
> **Response to Reviewer AmyZ [1/3]**
>
> Dear Reviewer AmyZ,
>
> We sincerely appreciate your thoughtful and constructive review. We have reviewed your comments carefully and provide detailed responses below. For clarity, we note that model names in our response are abbreviated as follows:
>
> * **Mistral 7B** = Mistral-7B-Instruct-v0.3, **Llama3 8B** = Llama-3.1-8B-Instruct, **Llama3 3B** = Llama-3.2-3B-Instruct, **Llama2 7B** = Llama-2-7b-chat-hf
>
> ---
>
> **[W1, Q4] The proposed method still relies on generating a large pool of synthetic samples, which may introduce significant cost ... Can the authors clarify the computational overhead of computing per-token log-likelihood differences compared to TransLoRA’s discriminator?**
>
> First, we would like to clarify that the primary contribution of our approach is token-level transfer, not synthetic data. With a particular focus on which tokens should be transferred and how those signals should be aligned, our work attempts to address the question of how the knowledge-transfer process itself should be designed. In fact, Table 3 of our paper contains experiments using external, non-synthetic datasets where the approach continues to perform strongly, further demonstrating that TiTok is not tied to synthetic data. We hope this clarifies that our method is neither limited nor dependent on synthetic data.
>
> In response to the question about the computational overhead, we would also like to point out the computational differences between our method and TransLoRA [1]. TransLoRA requires an additional phase for training a separate discriminator, which is computationally heavy and expensive. In contrast, TiTok does not train any additional models; instead, all steps are performed internally with the source model and its LoRA adapter. This design makes TiTok substantially lighter and more efficient. To provide rigorous empirical evidence, we conducted an end-to-end comparison of TransLoRA and TiTok on the BBH and News Headline benchmarks in the Mistral 7B → Llama3 8B transfer setting. Across both benchmarks, TiTok achieves roughly a 1.5×–2.5× reduction averagely in total compute time. The removal of the discriminator-training stage accounts for the majority of this improvement, while the per-token log-likelihood computation contributes only a minimal and manageable overhead.
> ​
>
> ### **BBH (250 samples × 27 tasks)**
>
> | Method        | Metric                          | Discriminator Training |     Data select |             |   Training | `​`    `​` TOTAL |
> | :------------ | :------------------------------ | ---------------------: | --------------: | -----------: | --------: | --------: |
> | **TransLoRA** | avg_total (sec/num_task)        |              279.33936 |        31.72915 |            — | 104.61636 | **415.68487** |
> |               | avg_sample (sec/sample)         |                0.63251 |         0.06346 |            — |   0.41847 |   **1.11444** |
> |               |               | **Contrastive  excess scoring** | **Data select** | **Token align** | **Training** | **TOTAL** |
> | **TiTok**     | avg_total (sec/num_task)        |              164.70895 |         0.01155 |      0.41080 | 102.02867 | **267.15997** |
> |               | avg_sample (sec/sample)         |                0.32942 |         0.00005 |      0.00054 |   0.40811 |   **0.73812** |
>
>
>
> ### **News Headline (200 samples × 30 users)**
>
> |        Method |                    Metric |         Discriminator Training |     Data select |                |   Training | `​`    `​` TOTAL |
> | ------------: | ------------------------: | -----------------------------: | --------------: | --------------: | -----------: | ------------: |
> | **TransLoRA** | avg_total (sec/num_users) |                      245.10085 |        25.39730 |               — |     58.71148 | **329.20963** |
> |               |   avg_sample (sec/sample) |                        0.61279 |         0.06358 |               — |      0.29356 |   **0.96993** |
> |               |                           | **Contrastive excess scoring** | **Data select** | **Token align** | **Training** |     **TOTAL** |
> | **TiTok**     | avg_total (sec/num_users) |                       81.31437 |         0.01863 |         0.89304 |     49.52768 | **131.75371** |
> |               |   avg_sample (sec/sample) |                        0.20355 |         0.00005 |         0.00447 |      0.24764 |   **0.45457** |
>
>
>
> We add this result and the corresponding discussion in Appendix I of the revised draft. We hope that this clarifies the computational efficiency of our method.

---

> ### Author Response · Authors · 2025-11-22
> **Response to Reviewer AmyZ [2/3]**
>
> **[W2] While token-level filtering is the paper’s key innovation, there is no qualitative or visual analysis demonstrating which tokens are retained or why they contribute to effective transfer.**
>
> Thank you for the suggestion. In response, we provide qualitative examples from the news headline task to illustrate how TiTok identifies the most informative tokens. The **bold** text below indicates the tokens selected by TiTok. Notably, these tokens are those with the highest contrastive excess score:
>
> **`<Qualitative example 1>`**
>
>
> Prompt:
> You are a news headline generator.
> Generate a headline for the following article.
> article: With a scalloped hem, it's a show-stopping piece that's both unique and affordable. Want more? Be sure to check out Stylelist on….
> headline:
>
> Output:
>
> Stylist'**:** **The** **Perfect** **White** Dress **For** Your **Wed**ding **(PHOT**OS) **<\/s>**
>
>
>
>
> **`<Qualitative example 2>`**
>
>
> Prompt:
> You are a news headline generator.
> Generate a headline for the following article.
> article: It’s hard to imagine that you would be interested in hearing about my favorite holiday tradition, but if you are, I’d be happy to share…..
> headline:
>
>
> Output:
>
> **How To** Make Your **Own DI**Y **Christmas** Or**naments (PH**OTOS) **<\/s>**
>
> Analytically, the selected tokens correspond to the beginnings of major semantic or structural units in the headline. These tokens carry the highest information value: they denote transitions, introduce core noun phrase components, or contain root words. Moreover, since the contrastive excess score is defined as the difference between the expert model (source LoRA + base) and the amateur model (base only), it pinpoints the tokens where the expert and amateur diverge most in their predictions and so include effective learning signals for target tasks. Taken together, this qualitative example demonstrates that TiTok consistently retains the tokens that are most important for providing structure and meaning. We have added these examples in Appendix Q of the revised draft.
>
>
> ---
>
> **[Q1] How sensitive is TITOK to the quality of the source LoRA? Would a poorly tuned LoRA produce misleading excess scores?**
>
>
> While it is true that TiTok uses the source LoRA as a reference, its effectiveness is not strongly dependent on the source LoRA’s absolute performance. Below, we additionally report the cases in Table 1 where the source LoRA underperforms the target model’s vanilla version, together with the source LoRA’s own scores. As shown in the table, TiTok still consistently improves the target model even when the source LoRA is weaker than the target baseline. This indicates that TiTok captures domain-specific signals that are embedded in certain tokens rather than extracting the entire capabilities of the source LoRA.
>
>
> | Model Transfer             |              Metric|  BBH        | Scholarly | Title |
> |---------------------------|--------------|------------:|----------:|-------:|
> |                           |              | Acc.       |     R-1   |   R-L |
> | **Llama3 3B**             | source lora  | 0.460      |     0.429 | 0.363 |
> | → **Llama3 8B**           | target base  | 0.462      |     0.444 | 0.378 |
> |                           | **TiTok (k=30%)** | **0.509** | **0.457** | **0.392** |
> |                           |              |            |           |       |
> | **Llama2 7B**             | source lora  | 0.359      |     0.431 | 0.367 |
> | → **Llama3 8B**           | target base  | 0.469      |     0.444 | 0.378 |
> |                           | **TiTok (k=70%)** | **0.510** | **0.461** | **0.404** |
>
> KD-based approaches [2,3], however, are highly dependent on the absolute capacity of the source LoRA. By making the target model follow the teacher's logits, a weak or poorly tuned source LoRA unavoidably transfers certain flaws, resulting in the minimal gains supported by the results in the above table. Meanwhile, TiTok avoids these deficiencies because it fundamentally differs in how it leverages the source LoRA. Instead of relying on the source LoRA’s logits globally, TiTok uses the source only to smartly identify informative token positions. As a result, a poorly tuned source LoRA does not mislead TiTok since it does not make the target model imitate the source LoRA’s behavior. Rather, it selectively retrieves the useful tokens using the source LoRA's domain knowledge.
>
> We add this result and the corresponding discussion in Appendix O of the revised draft.

---

> ### Author Response · Authors · 2025-11-22
> **Response to Reviewer AmyZ [3/3]**
>
> **[Q2] Could contrastive excess be used iteratively (e.g., to refine the LoRA itself) rather than just for one-shot transfer?**
>
> We thank you for posing an interesting perspective of our method. However, we would like to clarify that our primary goal in TiTok is knowledge transfer rather than self-refinement/improvement.
>
> Nonetheless, to address your interest in this direction, we conducted an experiment to examine whether TiTok can also be effective for self-refinement. We performed this experiment on the BBH benchmark in the Mistral 7B → Mistral 7B and the Mistral 7B → Llama3 8B transfer settings. After the initial transfer step, 1) we generated synthetic data using the target LoRA that had already learned TiTok’s transferred knowledge, 2) computed contrastive excess scores on the newly generated synthetic data using the original source LoRA and the its base model, and 3) trained a new target LoRA with the same hyperparameters used in the main paper. (e.g., k=0.7)
>
> | Model Transfer             | Method                          | Acc.  |
> |---------------------------|----------------------------------|-------|
> | **Mistral 7B**  | Vanilla                        | 0.397 |
> | → **Mistral 7B**    | Initial TiTok                    | 0.432 |
> |                           | **Iterative TiTok: self transfer** | **0.456** |
> |                           | Target expert                    | 0.460 |
> |                           |                                  |       |
> | **Mistral 7B**  | Vanilla                         | 0.469 |
> | → **Llama3 8B**                          | Initial TiTok                    | 0.482 |
> |                           | **Iterative TiTok: self transfer** | **0.510** |
> |                           | Target expert                    | 0.531 |
>
> As shown in the table, this iterative step yields additional improvement, suggesting that iterative refining is possible. We hypothesize that this is possible because the initial TiTok transfer increases the target model's familiarity with the task and partially aligns its internal representations with the source domain. As a result, the target model generates cleaner, more coherent synthetic data with fewer irrelevant patterns. This thereby enables the second iteration of TiTok to more accurately identify domain-informative tokens, which improves token selection and makes excess scoring more discriminative.
>
> However, we emphasize that iterative refining synthetic data cannot surpass a target model trained on real train data. By creating increasingly better synthetic samples, iteration can help narrow the gap, but it is still far from  a fully supervised target model trained on actual data.
>
> We have included this analysis in Appendix P of the revised draft.
>
> ---
>
> **[Q3] How does TITOK behave when transferring between models with different architectures (e.g., dense vs. mixture-of-experts backbones)?**
>
> In our paper, we have already considered transfer settings across various model sizes and architectures, thus demonstrating that TiTok is not limited to a single backbone family.
>
> Nonetheless, to satisfy your curiosity on this matter, we additionally conducted experiments on cross-architecture transfer from a dense model (Mistral 7B) to a mixture-of-experts (MoE) model (`Mixtral-8×7B-Instruct-v0.1`). All experiments were conducted under 4-bit quantization due to the large model sizes involved.
>
> | Model Transfer  | Metric              | Acc.  |
> |-----------------|---------------------|-------|
> | **Mistral 7B**  | Vanilla             | 0.454 |
> | → `Mixtral-8×7B-Instruct-v0.1` | **TiTok (k=70%)** | **0.463** |
>
> Analytically, TiTok outperforms the MoE target baseline, indicating that TiTok remains effective in the MoE setting.  This result suggests that TiTok can generalize to heterogeneous model architectures,  including dense → MoE transfer.
>
> We add this result and the corresponding discussion in Appendix J of the revised draft.
>
> ---
>
> [1] Wang et al., Trans-LoRA: towards data-free Transferable Parameter Efficient Finetuning, NeurIPS 2024
> [2] Wan et al., Knowledge Fusion of Large Language Models, ICLR 2024
> [3] Hinton et al., Distilling the Knowledge in a Neural Network., NeurIPS 2014 Workshop
>
> ---
>
> We hope our response adequately responds to your comments. If you have any further questions, please don't hesitate and let us know.
>
> Thank you very much,
> Authors

---

### Author Response · Authors · 2025-11-22
**General Response**

Dear reviewers and AC,

We are grateful for the thoughtful time and attention you have given to evaluating our work.

As the reviewers highlighted, TiTok  is a simple yet conceptually **novel framework for LoRA transfer (AmyZ, LDW6, g2oB)** that provides lightweight, **model-agnostic token-level knowledge extraction (AmyZ, YLn5)**. In addition to offering an efficient tokenizer-alignment method that even improves cross-family applicability, our work shows **consistent empirical benefits across a variety of tasks, models, and transfer settings (all reviewers)**.

We appreciate your constructive and valuable feedback on our manuscript.  We have carefully updated and improved the manuscript in response to the feedbacks , including the following additional discussions and experiments:

* Addition of “token-wise” to clarify the contrastive excess terminology (throughout the paper)
* Corrections on typos and improvements in sentence-level clarity (throughout the paper)
* Empirical analysis and explanation of tokenizer-alignment algorithm’s accuracy (Appendix A)
* Theoretical foundations and empirical validation of TiTok (Appendix B)
* Clarified comparison between TiTok and TransLoRA (Appendix D.3)
* Expanded Table 1 (=main table) with variance estimates and statistical details (Appendix F)
* Comprehensive sensitivity study on the k-percent token ratio (Appendix H)
* Discussion of computational overhead and efficiency of TiTok (Appendix I)
* Additional experiments on larger model scales (70B) and alternative architectures (e.g., Mixture-of-Experts) (Appendix J)
* Robustness analysis with respect to synthetic-data quality (Appendix M)
* Addressing synthetic-data quality and bias (Appendix N)
* Discussion of performance when the source LoRA is weak (Appendix O)
* Exploration of iterative self-refinement using TiTok (Appendix P)
* Qualitative examples and extended discussions (Appendix Q)

In the revised manuscript, these updates are temporarily highlighted in $\textbf{\color{green}green}$ for your convenience to check.

We sincerely believe that these updates may help us better deliver the benefits of the proposed TiTok to the ICLR community.

Sincerely,
Authors.

---

### Author Response · Authors · 2025-11-26
**A Gentle Invitation for Further Discussion**

Dear reviewers,

Once again, we would like to kindly express our appreciation for your thoughtful reviews and the time you have devoted to evaluating our submission.

If you have any additional thoughts or follow-up comments after reading our author response, we would be very grateful to hear them. Your insights during the discussion phase are extremely valuable and help us further clarify and strengthen the work.

Thank you very much for your time and engagement.

Kindest regards,
Authors

---

### Meta-Review · Area_Chair_EDaE · 2026-01-12

**Summary:**

Unlike prior work (e.g., TransLoRA) that relies on sequence-level transfer and auxiliary discriminators, TiTok introduces a "token-wise contrastive excess" mechanism to identify and transfer only the most informative tokens based on the log-likelihood ratio between the source LoRA and its base model. Reviewers appreciate the method for its novelty and efficiency. TiTok also demonstrates consistent and statistically significant gains across diverse transfer settings and demonstrated scalability to 70B parameter models and robustness to variations in synthetic data quality and prompt design. The authors provided a comprehensive rebuttal that resolved reviewer concerns regarding theoretical justification, scalability (adding 70B experiments), and baselines. The reviewers were unanimous in their final recommendation to accept.

**Reviewer Concerns:**

seems all the reviewers' concerns are addressed.

**Reviewer Scores:**

NA

---

### Decision · Program_Chairs · 2026-01-26

Accept (Poster)